# Tight Lower Bounds and Improved Convergence in Performative Prediction

**Pedram Khorsandi**
Mila, Quebec AI Institute
Université de Montréal

**Rushil Gupta**
Mila, Quebec AI Institute
Université de Montréal

**Mehrnaz Mofakhami**
Mila, Quebec AI Institute
Université de Montréal

**Simon Lacoste-Julien**[*]
Mila, Quebec AI Institute
Université de Montréal
Canada CIFAR AI Chair

**Gauthier Gidel**[*]
Mila, Quebec AI Institute
Université de Montréal
Canada CIFAR AI Chair

## Abstract

Performative prediction is a framework accounting for the shift in the data distribution induced by the prediction of a model deployed in the real world. Ensuring convergence to a stable solution—one at which the post-deployment data distribution no longer changes—is crucial in settings where model predictions can influence future data. This paper, for the first time, extends the Repeated Risk Minimization (RRM) algorithm class by utilizing historical datasets from previous retraining snapshots, yielding a class of algorithms that we call Affine Risk Minimizers that converges to a performatively stable point for a broader class of problems. We introduce a new upper bound for methods that use only the final iteration of the dataset and prove for the first time the tightness of both this new bound and the previous existing bounds within the same regime. We also prove that our new algorithm class can surpass the lower bound for standard RRM, thus breaking the prior lower bound, and empirically observe faster convergence to the stable point on various performative prediction benchmarks. We offer at the same time the first lower bound analysis for RRM within the class of Affine Risk Minimizers, quantifying the potential improvements in convergence speed that could be achieved with other variants in our scheme.

## 1 Introduction

Decision-making systems are increasingly integral to critical judgments in sectors such as public policy [Fire and Guestrin, 2019], healthcare [Bevan and Hood, 2006], and education [Nichols and Berliner, 2007]. However, as these systems become more reliant on quantitative indicators, they become vulnerable to the effects described by Goodhart's Law: "When a measure becomes a target, it ceases to be a good measure" [Goodhart, 1984]. This principle is particularly relevant when predictive models not only forecast outcomes but also influence the behavior of individuals and organizations, leading to performative effects that can subvert the original goals of these systems.

For example, in environmental regulation, companies might manipulate emissions data to meet regulatory targets without truly reducing pollution, thus distorting the intended environmental protection efforts [Fowlie et al., 2012]. In healthcare, hospitals may modify patient care practices to improve performance metrics, potentially prioritizing score improvements over actual patient health outcomes [Bevan and Hood, 2006]. Similarly, in education, the emphasis on standardized test scores can lead

---

[*]Equal advising. Correspondence to Pedram Khorsandi: `<pedram.khorsandi@mila.quebec>`

39th Conference on Neural Information Processing Systems (NeurIPS 2025).

schools to focus narrowly on test preparation, compromising the broader educational experience [Nichols and Berliner, 2007]. These examples demonstrate how decision systems, when overly focused on specific indicators, can be manipulated, resulting in the corruption of the very processes they aim to enhance.

Given these challenges, it is essential to develop predictive models that are not only accurate but also robust against the performative shifts they may provoke. The work by Perdomo et al. [2020] addresses this challenge within the framework of *Repeated Risk Minimization (RRM)*, where they explore the dynamics of model retraining in the presence of performative feedback loops. In their approach, the authors propose an iterative method that adjusts the predictive model based on the distributional shifts caused by prior model deployments, aiming to stabilize the model performance despite the continuous evolution of the underlying data distribution. By characterizing the convergence properties of their method, they provide a theoretical guarantee for the stability of the model at a performative equilibrium.

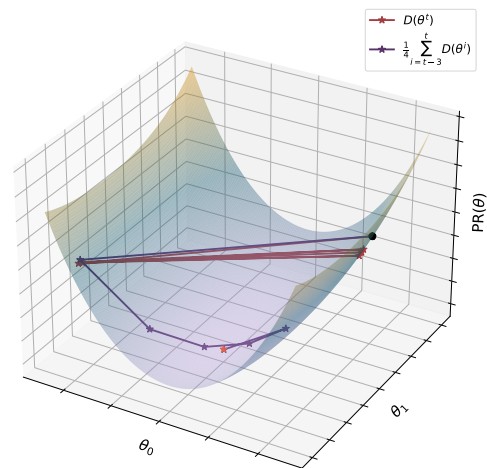

Figure 1: An example showing that using older snapshots (purple) speeds up convergence to the stable point (orange star) compared to only the latest snapshot (red). The implementation is provided in the code.

Our work extends this framework by leveraging the datasets collected at each snapshot during the retraining process, introducing a new class of algorithms called *Affine Risk Minimizers* (ARM). By utilizing historical data from previous updates, we show that it is possible to converge to a stable point for a broader class of problems that were previously unsolvable, extending beyond the bounds established in prior analyses [Mofakhami et al., 2023]. We derive a new upper bound under less restrictive assumptions than Mofakhami et al. [2023] and provide the first tightness analysis for the framework in Perdomo et al. [2020] as well as for our newly established rate. Our method, which incorporates historical datasets, demonstrates superior convergence properties both theoretically and experimentally.

Converging to a stable point is essential in decision-dependent learning systems. Without stability, iterative retraining may lead to persistent fluctuations, preventing reliable long-term predictions. Prior work has examined convergence rates and the range of problem classes in which iterative schemes can achieve stability [Li and Wai, 2024, Narang et al., 2024, Perdomo et al., 2020]. This paper provides the first analytical techniques for examining the tightness of the upper bounds for these methods. In addition, we show that the ARM framework ensures convergence to a stable solution for a broader class of problems, mitigating the limitations of existing approaches.

**Contributions.** ① We establish a new upper bound, enhancing the convergence rate of RRM under less restrictive conditions than [Mofakhami et al., 2023]; ② We establish the tightness of the analysis in both the framework proposed by [Perdomo et al., 2020] and our modification of the framework from Mofakhami et al. [2023]; ③ We introduce a new class of algorithms, named Affine Risk Minimizers (ARM), that provides convergence for a wider class of problems by utilizing linear combinations of datasets from earlier training snapshots; ④ We provide both theoretical and experimental enhancements, showcasing scenarios where ARM improves convergence; ⑤ Finally, we present the first lower bound techniques for iterative retraining schemes and apply it to both Perdomo et al. [2020] and our modified framework to establish **theoretical lower bounds** for ARM, detailing the maximum potential improvement in convergence rates achievable through the use of past datasets.

## 2 Related Work

Performative prediction introduces a framework for learning under decision-dependent data [Perdomo et al., 2020], and has been widely studied in various aspects, from stochastic optimization methods to

find stable classifiers [Li and Wai, 2022, Mendler-Dünner et al., 2020] to approaches that focus on performative optimal solutions, the minimizer of performative risk [Miller et al., 2021, Jagadeesan et al., 2022, Lin and Zrnic, 2024]. In this work, we focus our analysis on performative stable solutions, whose deployment removes the need for repeated retraining in changing environments [Perdomo et al., 2020, Mendler-Dünner et al., 2020, Jagadeesan et al., 2021, Brown et al., 2022, Mofakhami et al., 2023].

One of the main applications of this framework is strategic classification [Hardt et al., 2016] which involves deploying a classifier interacting with agents who strategically adapt their features to alter the classifier's predictions and achieve their favorable outcomes. Strategic Classification has been widely used in the literature of performative prediction [Perdomo et al., 2020, Mendler-Dünner et al., 2020, Miller et al., 2021, Hardt et al., 2022, Mofakhami et al., 2023, Narang et al., 2024, Góis et al., 2025], and we adopt this setting in our experiments to empirically demonstrate our theoretical contributions.

Prior work in performative prediction either assumes the data distribution is a function of the parameters modeled as $\mathcal{D}(\theta)$ [Perdomo et al., 2020, Izzo et al., 2021, Drusvyatskiy and Xiao, 2022, Dong et al., 2023], or more realistically dependent on the predictions as in $\mathcal{D}(f_\theta)$ [Mofakhami et al., 2023, Mendler-Dünner et al., 2022]. Although existing work only assumes one of these settings, our work adheres to both, by providing a tightness analysis of the rates proposed in Perdomo et al. [2020] and Mofakhami et al. [2023] and showcasing scenarios where we can provide an improved convergence by considering the history of distributions. To the best of our knowledge, we are first to provide a lower bound on the convergence rates achievable using any such affine combination of previous snapshots.

For the assumptions, we adopt those in Mofakhami et al. [2023] and Perdomo et al. [2020]. While several works pursue performatively optimal solutions by assuming convexity over the performative risk itself, recent efforts such as Zheng et al. [2024], Cyffers et al. [2024] relax these requirements by considering non-convex objectives or weaker regularity conditions. In contrast, our analysis imposes no convexity assumption on the performative risk, relying only on convexity of the loss function to establish convergence guarantees.

Most related to our idea of using previous distributions are works that study gradually shifting environments considering history dependence [Brown et al., 2022, Li and Wai, 2022, Rank et al., 2024]. Brown et al. [2022] brought up the notion of stateful performative prediction studying problems where the distribution depends on the classifier and the previous state of the population. This is modeled by a transition function that is fixed but a priori unknown and they show that by imposing a Lipschitz continuity assumption similar to $\epsilon$-sensitivity to the transition map, they can prove the convergence of RRM to an equilibrium distribution-classifier pair. In our work, we consider a specific dependence on history, by using an affine combination of previous distributions, and show that this can lead to an improved convergence than prior work without imposing any additional assumption.

## 3 Performative Stability and RRM

In the context of performative prediction, two primary frameworks are commonly used to address the challenge of shifting distributions due to the model's influence: Repeated Risk Minimization (RRM) and Repeated Gradient Descent (RGD). Throughout this paper, we focus on RRM.

RRM iteratively retrains the model on the distribution it induces until it converges to a performatively stable classifier. Formally, consider a model $f_\theta \in \mathcal{F}$ with parameters $\theta \in \Theta$, and a distribution $D(f_\theta)$ that depends on these parameters. The performative risk is defined as:

$$\text{PR}(\theta) = \mathbb{E}_{z \sim D(f_\theta)} \left[ \ell(f_\theta(x), y) \right] \tag{1}$$

where $\ell(f_\theta(x), y)$ is the loss function for a data point $z = (x, y)$. A classifier is performatively stable if it minimizes the performative risk on the distribution it induces:

$$\theta_{\text{PS}} = \arg\min_{\theta \in \Theta} \mathbb{E}_{z \sim D(f_{\theta_{\text{PS}}})} \left[ \ell(f_\theta(x), y) \right] \tag{2}$$

The RRM framework updates the model parameters by solving:

$$\theta^{t+1} = \arg\min_{\theta \in \Theta} \mathbb{E}_{z \sim D(f_{\theta^t})} \left[ \ell(f_\theta(x), y) \right] \tag{3}$$

until convergence, i.e., $\theta^{t+1} \approx \theta^t$.

# 4 Improved Rates and Optimality of Analysis

Both Perdomo et al. [2020] and Mofakhami et al. [2023] derive convergence rates for RRM under distinct assumptions. The assumptions made in these studies reflect the sensitivity of the distribution map $\mathcal{D}(.)$ to changes in the model and the structural properties of the loss function. Specifically, Perdomo et al. [2020] focuses on Wasserstein-based sensitivity and convexity with respect to the model parameters, while Mofakhami et al. [2023] introduces a framework with Pearson $\chi^2$-based sensitivity and strong convexity with respect to the predictions. Building on these foundations and motivated by Mofakhami et al. [2023], we now outline the assumptions for our framework, which departs from Mofakhami et al. [2023] only in Assumption 1 below.

**Assumption 1** $\epsilon$-*sensitivity w.r.t. Pearson $\chi^2$ divergence: The distribution map $\mathcal{D}(f_\theta)$, with pdf $p_{f_\theta}$, maintains $\epsilon$-sensitivity with respect to Pearson $\chi^2$ divergence. Formally, for any $f_\theta, f_{\theta'} \in \mathcal{F}$:*

$$\chi^2(\mathcal{D}(f_{\theta'}), \mathcal{D}(f_\theta)) \le \epsilon \|f_\theta - f_{\theta'}\|_{f_\theta}^2, \tag{4}$$

*where*

$$\|f_\theta - f_{\theta'}\|_{f_{\theta*}}^2 := \int \|f_\theta(x) - f_{\theta'}(x)\|^2 p_{f_{\theta*}}(x) dx \qquad \forall f_{\theta*} \in \mathcal{F}, \tag{5}$$

*and*

$$\chi^2(\mathcal{D}(f_{\theta'}), \mathcal{D}(f_\theta)) := \int \frac{\left(p_{f_{\theta'}}(z) - p_{f_\theta}(z)\right)^2}{p_{f_\theta}(z)} dz. \tag{6}$$

This assumption, inspired by prior work, is Lipschitz continuity on $D(.)$, implying that if two models with similar prediction functions are deployed, the distributions they induce should also be similar.

**Assumption 2** *Norm equivalency: The distribution map $\mathcal{D}(f_\theta)$ satisfies norm equivalency with parameters $C \ge 1$ and $c \le C$. For all $f_\theta, f_{\theta'}, f_{\theta*} \in \mathcal{F}$:*

$$c\|f_\theta - f_{\theta'}\|_{f_{\theta*}}^2 \le \|f_\theta - f_{\theta'}\|^2 \le C\|f_\theta - f_{\theta'}\|_{f_{\theta*}}^2, \tag{7}$$

*where*

$$\|f_\theta - f_{\theta'}\|^2 = \int \|f_\theta(x) - f_{\theta'}(x)\|^2 p(x) dx, \tag{8}$$

*and $p(x)$ is the initial distribution, referred to as the base distribution.*

The base distribution $p(x)$, following prior formulations in the literature [Perdomo et al., 2020, Mofakhami et al., 2023, Brown et al., 2022], corresponds to the pre-deployment data distribution. This interpretation of $p(x)$ as an intervention-free or organic distribution is also consistent with other areas of the literature. Schnabel et al. [2016] demonstrate this in the context of recommender systems, where unbiased test sets like Yahoo! R3 are constructed to reflect user behavior prior to any algorithmic influence.

This assumption holds whenever the distribution map satisfies the bounded density ratio property, i.e., $c\,p_{f_\theta}(x) \le p(x) \le C\,p_{f_\theta}(x)$ for all $f_\theta \in \mathcal{F}$. In such cases, one can define small constants $c := \inf_{\theta,x} \frac{p_{f_\theta}(x)}{p(x)}$ and $C := \sup_{\theta,x} \frac{p_{f_\theta}(x)}{p(x)}$. We measure these constants in Appendix I for our experimental setup in Section 7.

**Assumption 3** *Strong convexity w.r.t. predictions: The loss function $\hat{y} \mapsto \ell(\hat{y}, y)$ is $\gamma$-strongly convex. For any differentiable function $\ell$, and for all $y, \hat{y}_1, \hat{y}_2 \in \mathcal{Y}$:*

$$\ell(\hat{y}_1, y) \ge \ell(\hat{y}_2, y) + (\hat{y}_1 - \hat{y}_2)^\top \nabla_{\hat{y}} \ell(\hat{y}_2, y) + \frac{\gamma}{2} \|\hat{y}_1 - \hat{y}_2\|^2.$$

**Assumption 4** *Bounded gradient norm: The loss function $\ell(f_\theta(x), y)$ has a bounded gradient norm, with an upper bound $M = \sup_{x,y,\theta} \|\nabla_{\hat{y}} \ell(f_\theta(x), y)\|$.*

Building upon Mofakhami et al. [2023], we introduce a new theorem that demonstrates faster linear convergence for RRM, showing that stability can be achieved under less restrictive conditions.

**Theorem 1 (RRM convergence modified Mofakhami's framework)** *Suppose the loss $\ell(f_\theta(x), y)$ is $\gamma$-strongly convex with respect to $f_\theta(x)$ (A3) and that the gradient norm with respect to $f_\theta(x)$ is bounded by $M = \sup_{x,y,\theta} \|\nabla_{\hat{y}} \ell(f_\theta(x), y)\|$ (A4). Let the distribution map $\mathcal{D}(\cdot)$ be $\epsilon$-sensitive with respect to the Pearson $\chi^2$ divergence (A1), satisfy norm equivalency with parameters $C \geq 1$ and $c \leq C$ (A2), and the function space $\mathcal{F}$ be convex and compact under the norm $\|\cdot\|$.*

*Then, for $G(\theta^t) = \arg\min_{\theta \in \Theta} \mathbb{E}_{z \sim \mathcal{D}(f_{\theta^t})} \ell(f_\theta(x), y)$, with $z = (x, y)$, we have[2]:*

$$\|f_{G(\theta)} - f_{G(\theta')}\|_{f_\theta} \leq \frac{\sqrt{\epsilon}M}{\gamma} \|f_\theta - f_{\theta'}\|_{f_\theta}.$$

*By the Schauder fixed-point theorem, a stable classifier $f_{\theta_{PS}}$ exists, and if $\frac{\sqrt{\epsilon}M}{\gamma} < 1$, RRM converges to a unique stable point $f_{\theta_{PS}}$ at a linear rate:*

$$\|f_{\theta^t} - f_{\theta_{PS}}\|_{f_{\theta_{PS}}} \leq \left(\frac{\sqrt{\epsilon}M}{\gamma}\right)^t \|f_{\theta_0} - f_{\theta_{PS}}\|_{f_{\theta_{PS}}}.$$

This shows that RRM achieves linear convergence to a stable classifier, provided that $\frac{\sqrt{\epsilon}M}{\gamma} < 1$, ensuring that the mapping is contractive and guarantees convergence. This result improves upon Mofakhami et al. [2023] by eliminating the constant $C$ from the rate, as defined in Assumption 2. Additionally, this approach can achieve improved rates of convergence, as discussed in Theorem 8, where we show how the new definition of $\epsilon$-sensitivity leads to faster convergence.

Despite these improvements, the following theorem establishes for the first time a lower bound under the given assumptions, indicating that the convergence rate cannot be further improved without additional conditions:

**Theorem 2 (Tight lower bound modified Mofakhami's framework)** *Suppose that Assumptions 1-4 hold, with parameters $\epsilon$, $M$, and $\gamma$ such that $\frac{\sqrt{\epsilon}M}{\gamma} \leq 1$. Under these conditions, there exists a problem instance such that, utilizing RRM, the following holds:*

$$\|f_{\theta^t} - f_{\theta_{PS}}\|_{f_{\theta_{PS}}} = \Omega\left(\left(\frac{\sqrt{\epsilon}M}{\gamma}\right)^t\right). \tag{9}$$

*If instead $\frac{\sqrt{\epsilon}M}{\gamma} > 1$, the bound is $\Omega(1)$, indicating non-convergence.*

This result establishes the tightness of the convergence rate under the specific assumptions outlined earlier, demonstrating that the bound cannot be improved without imposing more restrictive assumptions. The full proof of this theorem can be found in Appendix D. A similar tightness analysis for the framework proposed by Perdomo et al. [2020] is provided in the following section, confirming that both frameworks achieve optimal convergence guarantees given their respective conditions.

## 4.1 Tightness Analysis in Perdomo et al. [2020]'s Framework

In their work, Perdomo et al. [2020] make a set of assumptions that differ from Assumption 1-4. Their $\epsilon$-sensitivity assumption is with respect to the Wasserstein distance, and their strong convexity assumption is with respect to the parameters. Formally, they make the following set of assumptions to show the convergence of RRM

**Assumption 5** *The distribution map $\theta \mapsto D(\theta)$ is $\epsilon$-sensitive w.r.t $\mathcal{W}_1$:*

$$\mathcal{W}_1(\mathcal{D}(\theta), \mathcal{D}(\theta')) \leq \epsilon \|\theta - \theta'\|_2,$$

*the loss function $\theta \mapsto \ell(z : \theta)$ of the performative risk (1) is $\gamma$-strongly convex for any $z \in \mathcal{Z}$ and $z \mapsto \nabla_\theta \ell(z : \theta)$ is $\beta$-Lipschitz for any $\theta \in \Theta$.*

---

[2]Throughout this work, whenever we refer to $f_{G(\theta)}$, it denotes $f_{\hat{\theta}}$, where $\hat{\theta} \in G(\theta)$.

Under these assumptions and for $\frac{\beta\epsilon}{\gamma} < 1$, Perdomo et al. [2020] showed that RRM does converge to a performatively stable point at a rate:[3]

$$\|\theta^t - \theta_{\text{PS}}\| \leq \left(\frac{\beta\epsilon}{\gamma}\right)^t \|\theta_0 - \theta_{\text{PS}}\|. \tag{10}$$

We note that $\epsilon$-sensitivity with respect to the $\chi^2$ divergence (Assumption 1) is generally a stronger condition than $\epsilon$-sensitivity with respect to the Wasserstein distance $\mathcal{W}_1$, particularly when the input space has small diameter [Mofakhami et al., 2023]. While $\chi^2$ sensitivity implies tighter control over the induced distributional shifts, the two notions are not equivalent, and one does not necessarily imply the other. Hence, each framework is analyzed under its respective assumption.

**Theorem 3 (Tight lower bound Perdomo's framework)** *There exists a problem instance and an initialization $\theta_0$ following Assumption 5 such that employing RRM, we have:*

$$\|\theta^t - \theta_{PS}\| = \Omega\left(\left(\frac{\epsilon\beta}{\gamma}\right)^t \|\theta_0 - \theta_{PS}\|\right). \tag{11}$$

The proof uses a quadratic loss $\ell(z, \theta) = \frac{\gamma}{2}\|\theta - \frac{\beta}{\gamma}z\|^2$ and a performative distribution $z \sim \mathcal{N}(\epsilon\theta, \sigma^2)$ satisfying $\epsilon$-sensitivity under $\mathcal{W}_1$. Hence, the RRM update $\theta^{t+1} = \epsilon\frac{\beta}{\gamma}\theta^t$ matches the contraction factor $(\epsilon\frac{\beta}{\gamma})^t$ in Perdomo et al. [2020], confirming the bound's tightness. More detailed proof of this result is provided in Appendix C.

Our theorems show that given the assumptions in either framework, the convergence rate for RRM reaches a fundamental lower bound. This implies that further improvements in convergence speed would require either more restrictive assumptions or a novel optimization framework.

In the next section, we present, for the first time, an approach that breaks the RRM algorithm class by exploiting data from earlier training snapshots, and thereby surpasses the established lower bound, providing improved convergence guarantees.

## 5 Usage of Old Snapshots: Affine Risk Minimizers

Instead of relying solely on the current data distribution induced by $D(f_{\theta^t})$, we leverage datasets from previous training snapshots $\{D(f_{\theta^i})\}_{i=0}^{t-1}$. The new scheme optimizes model parameters over an aggregated distribution:

$$\theta^{t+1} = \arg\min_{\theta\in\Theta} \mathbb{E}_{(x, y)\sim D_t} [\ell(f_\theta(x), y)] \tag{12}$$

where $D_t$ is an affine combination of previous distributions, formulated as:

$$D_t = \sum_{i=0}^{t-1} \alpha_i^{(t)} D(f_{\theta^i}), \quad \text{s.t.} \quad \sum_{i=0}^{t-1} \alpha_i^{(t)} = 1 \tag{13}$$

We refer to this class of algorithms as *Affine Risk Minimizers*. As demonstrated in Appendix A (Lemma 9), the set of stable points for this class of algorithms coincides with those obtained through standard RRM. The following lemma formalizes the convergence of ARM under the stated assumptions, using only the average of the final two training snapshots.

**Lemma 1 (2-Snapshots ARM recurrence)** *Consider the class of problems for which Assumptions 1-4 are satisfied, and let the distribution map $\mathcal{D}(.)$ be $\frac{\epsilon}{C}$-sensitive with respect to the base distribution within the convex function space $\mathcal{F}$. Formally, for any $f_\theta, f_{\theta'} \in \mathcal{F}$,*

$$\chi^2(\mathcal{D}(f_{\theta'}), \mathcal{D}(f_\theta)) \leq \frac{\epsilon}{C}\|f_\theta - f_{\theta'}\|^2,$$

---

[3]Note that if $\frac{\beta\epsilon}{\gamma} \geq 1$ the convergence rate is vacuous. In that case, a performatively stable point may not even exist.

where $\|f_\theta - f_{\theta'}\|^2$ is defined in Equation 8. The distribution at iteration $t$ is given by

$$D_t = \frac{1}{2}D(f_{\theta^t}) + \frac{1}{2}D(f_{\theta^{t-1}}). \tag{14}$$

Under these conditions, the following convergence property holds for the iterative sequence generated by Equation 12:

$$\|f_{\theta^{t+1}} - f_{\theta^t}\| = \left( \frac{\sqrt{3}}{2} \frac{\sqrt{\epsilon}M}{\gamma} \right) m_t, \tag{15}$$

where $m_t = \max\{\|f_{\theta^t} - f_{\theta^{t-1}}\|, \|f_{\theta^{t-1}} - f_{\theta^{t-2}}\|\}$.

The problem class defined here aligns with that in Theorem 2 for the case $C \approx 1$. Now, the following theorem provides theoretical evidence of improved convergence, which will be further supported by experiments in Section 7.

**Theorem 4 (2-Snapshots ARM convergence)** *If* $\left( \frac{\sqrt{3}}{2} \frac{\sqrt{\epsilon}M}{\gamma} \right) < 1$, *the sequence described in Lemma 1 forms a Cauchy sequence, converging to a stable point.*

Relaxing the earlier condition $\frac{\sqrt{\epsilon}M}{\gamma} \leq 1$ to the threshold $\frac{2}{\sqrt{3}} \approx 1.155$, this theorem demonstrates a modest but tangible improvement allowing ARM to breach the lower bound of Theorem 2 and shows that when $\frac{\sqrt{\epsilon}M}{\gamma} < \frac{2}{\sqrt{3}} \approx 1.155$ with $C \approx 1$, convergence to a stable point remains possible under the same conditions, in contrast to standard RRM, which does not converge. A detailed proof of this theorem, along with Lemma 1, is provided in Appendix E, where we show that the algorithm generates a Cauchy sequence and converges to the stable point. We prove the convergence for schemes that use the average of the last $n$ snapshots, for any $n$, but the best result is obtained for $n = 2$ so far.

In the following section, we explore the lower bound for the convergence rates achievable using any affine combination of previous snapshots.

# 6 Lower Bounds for Affine Risk Minimizers

We established the potential for convergence across a wider class of problems using ARMs. This prompts the question of how much the convergence class can be improved, which we address in this section.

We propose the first distinct lower bounds for the framework described in Section 4 and that of Perdomo et al. [2020] for the class of Affine Risk Minimizers. The lower bound for our framework is presented in this section, while the corresponding result for Perdomo et al. [2020] is detailed in the following.

**Theorem 5 (ARM lower bound modified Mofakhami's framework)** *Suppose that Assumptions 1-4 hold. Then, there exists a problem instance in this regime, and for any algorithm in the Affine Risk Minimizers class, such that:*

$$\|f_{\theta^t} - f_{\theta_{PS}}\|_{f_{\theta_{PS}}} = \Omega \left( \left( \frac{1}{\frac{1}{e} + 2} \frac{\sqrt{\epsilon}M}{\gamma} \right)^t \right). \tag{16}$$

This demonstrates that the convergence rate for the class of problems satisfying Assumptions 1-4 cannot exceed the given lower bound.

## 6.1 Lower Bound with Perdomo et al. [2020]'s Assumption

We show that the convergence rate for RRM provided in Equation 10 is optimal among the class of *Affine Risk Minimizers* up to a factor 2.

**Theorem 6 (ARM lower bound Perdomo's framework)** *There exists a problem instance and an initialization $\theta_0$ following Assumption 5 such that for any algorithm in the Affine Risk Minimizers class, we have:*

$$\|\theta^t - \theta_{PS}\| = \Omega\left(\left(\frac{\epsilon\beta}{2\gamma}\right)^t \|\theta_0 - \theta_{PS}\|\right). \tag{17}$$

**Proof Sketch.** The proofs of Theorems 5 and 6 are inspired by the idea of introducing a new dimension at each iteration by Nesterov's lower bound for convex smooth functions [Nesterov, 2014]. We construct a problem instance satisfying Assumption 5 and derive the iteration dynamics of RRM to establish a lower bound on its convergence rate.

We introduce a structured transformation of the parameter space, different from the one introduced in Nesterov [2014], using the matrix

$$A = \begin{bmatrix} 1 & 0 & 0 & \dots & 0 \\ 1 & 1 & 0 & \dots & 0 \\ 0 & 1 & 1 & \dots & 0 \\ \vdots & \vdots & \ddots & \ddots & \vdots \\ 0 & \dots & 0 & 1 & 1 \end{bmatrix} \in \mathbb{R}^{d \times d}.$$

This matrix ensures that if a vector $v$ is in the span of $\{e_1, ..., e_i\}$, where $e_i$ is the standard basis vector in $\mathbb{R}^d$, then $Av$ extends the span to include $e_{i+1}$. This property allows us to control the iterative exploration of dimensions in the distribution mapping.

We define the loss function as $\ell(f_\theta(x), y) = \frac{\gamma}{2}\|\theta - \frac{\beta}{\gamma}z\|^2$, and set the distribution map to $D(\theta) = \mathcal{N}\left(\frac{\epsilon}{2}A\theta + e_1, \sigma^2\right)$. This choice ensures that each RRM iteration follows the update rule

$$\theta^{t+1} = \frac{\beta}{\gamma}\left(\frac{\epsilon}{2}A\theta^t + e_1\right).$$

From Lemma 9 (Appendix A), we know that any Affine Risk Minimizer converges to the same set of stable points as RRM. Since the problem instance we constructed has a unique stable point, it remains to explicitly compute this point and derive a lower bound by summing the contributions from undiscovered dimensions, completing the proof. A detailed proof is provided in Appendix F. □

To further illustrate this, Figure 2 provides empirical evidence supporting the theoretical lower bound derived for Perdomo et al. [2020]. The figure shows the convergence of $\|\theta - \theta_{PS}\|$ over multiple iterations for various combinations of previous snapshots. As indicated by the dotted line, the lower bound is never violated, demonstrating that the theoretical result holds in practice. The experimental setup for these results is also detailed in Appendix F. We also extend our result to a more general case for ARM in the following theorem (with proof in Appendix F.1).

**Theorem 7 (Proximal ARM lower bound)** *Let Assumption 5 hold and generate the iterates via*

$$\theta^{t+1} = \arg\min_{\theta \in \Theta} \mathbb{E}_{(x,y) \sim D_t}\left[\ell(f_\theta(x), y)\right] + \frac{\lambda}{2}\|\theta - \theta^t\|^2, \tag{18}$$

*where $D_t$ is any mixture distribution defined in Equation 13. Then there exists a problem instance and an initialization $\theta_0$ such that every algorithm in the Affine Risk Minimizers class satisfies*

$$\|\theta^t - \theta_{PS}\| = \Omega\left(\left(\frac{\epsilon\beta}{2\gamma}\right)^t \|\theta_0 - \theta_{PS}\|\right). \tag{19}$$

The proximal ARM update in Equation 18 is the ARM version of the proximal formulation of RRM introduced by Drusvyatskiy and Xiao [2022]. In fact, the lower bound of Theorem 7 continues to hold for the standard RRM method, since RRM corresponds to the special case of ARM in which the mixture distribution $D_t$ places all its weight on the most recent iterate. Thus, this result demonstrates that the proof technique we have developed for establishing exponential lower bounds is not limited to the ARM family but extends naturally to other iterative decision-dependent optimization procedures. In the supplementary material, we also present convergence rates for Proximal RRM for the first time on the framework of Perdomo et al. [2020], filling a gap in the literature, though these rates offer no improvement over standard RRM.

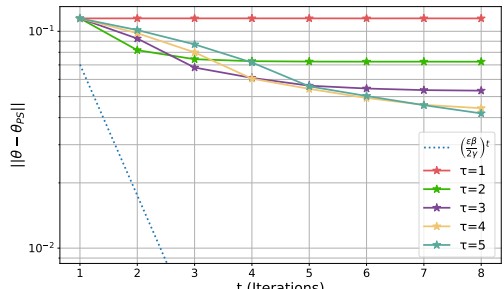
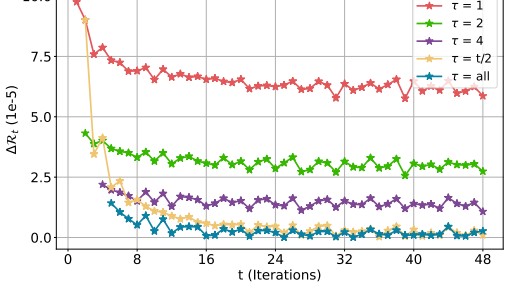

Figure 2: Convergence of $\|\theta^t - \theta_{\text{PS}}\|$ over iterations $t$ for different values of $\tau$, which defines the aggregation of datasets from training snapshots: $D_t = \sum_{i=t-\tau+1}^{t} \frac{1}{\tau} D(\theta_i)$. The dotted line shows our lower bound from Perdomo et al. [2020], with $\epsilon = 2.49$, $\beta = 1$, and $\gamma = 5.0$. The experiment, consistent across all methods, validates the bound by showing that $\|\theta^t - \theta_{\text{PS}}\|$ does not drop below it, supporting our theory.

Figure 3: Loss shift due to performativity for the credit-scoring environment. To accurately measure Performative Risk, we average over 500 runs per method. Increasing the aggregation window $\tau$ ($1 \to 2 \to 4 \to t/2 \to$ all) reduces the loss shifts and, consequently, reaches the stable point faster.

## 7 Experiments

We conduct experiments in two semi-synthetic environments to evaluate whether aggregating past snapshots improves convergence to the performatively stable point. We present an empirical comparison of different averaging windows for prior snapshots. At each time step $t$, we form $D_t$ by aggregating the datasets from the training snapshots as

$$D_t = \frac{1}{\tau} \sum_{i=t-\tau+1}^{t} D(f_{\theta^i}),$$

where we compare methods using various values of $\tau$, including $\tau = 1, 2, 4, \frac{t}{2}$, and 'all' (which includes all snapshots up to time $t$).

We first discuss our evaluation metric, then present detailed case studies on the credit scoring environment Mofakhami et al. [2023] (Section 7.1) and the rideshare markets Narang et al. [2024] (Appendix J).

**Evaluation Metric.** Throughout our experiments, we focus on changes in loss as a result of performativity. We define $\Delta\mathcal{R}_t$, i.e. the loss shift due to performativity at time $t$, as the absolute difference in loss observed by a model before and after the data distribution has changed due to performative effects while keeping the model's state constant.

$$\Delta\mathcal{R}_t = |\mathbb{E}_{z \sim \mathcal{D}(f_{\theta^t})}[\ell(f_{\theta^t}(x), y)] - \mathbb{E}_{z \sim \mathcal{D}(f_{\theta^{t-1}})}[\ell(f_{\theta^t}(x), y)]| \tag{20}$$

This metric allows for clearer comparisons between methods by minimizing overlap in the plots, unlike the performative risk (Equation 1).

### 7.1 Credit Scoring

**Setup.** Inspired by Mofakhami et al. [2023], we use the *Resample-if-Rejected (RIR)* procedure to model distribution shifts in a controlled experimental setting. This methodology involves users strategically altering their data to influence the classification outcome.

Let us consider a base distribution with probability density function $p$ and a function $g : f_\theta(x) \mapsto g(f_\theta(x))$ indicating the probability of rejection based on the prediction $f_\theta(x) \in \mathbb{R}$. The modified distribution $p_{f_\theta}$, under the *RIR* mechanism, evolves as follows:

- Sample $x$ from $p$.
- With probability $1 - g(f_\theta(x))$, accept and output $x$. Otherwise, resample from $p$.

Our data comes from Kaggle's *Give Me Some Credit* dataset[4], which includes features $x \in \mathbb{R}^{11}$ and labels $y \in \{0, 1\}$, where $y = 1$ indicates a defaulting applicant. We partition the features into two sets: strategic and non-strategic. We assume independence between strategic and non-strategic features. While non-strategic features remain fixed, the strategic features are resampled using the *RIR* procedure with a rejection probability $g(f_\theta(x)) = f_\theta(x) + \delta$. We use a scaled sigmoid function after the second layer. This scales $f_\theta(x)$ to the interval $[0, 1 - \delta]$, ensuring that $g(f_\theta(x)) \in [\delta, 1]$ remains a valid probability. Further implementation details are available in Appendix I.

**Theorem 8** *Let $f_\theta(x) \in [0, 1 - \delta]$ for all $\theta \in \Theta$, where $0 < \delta < 1$ is fixed. Then, for $g(f_\theta(x)) = f_\theta(x) + \delta$, RIR is $\epsilon$-sensitive as defined in Assumption 1 with $\epsilon = \mathcal{O}(\delta^{-\frac{3}{2}})$.*

This result provides an example where our rate surpasses the rate previously derived in Mofakhami et al. [2023] ($\mathcal{O}(\delta^{-2})$ within the same framework). In addition, Mofakhami et al. [2023] derived the remaining constants in the rate for this setup, and these constants remain unchanged in our result. Furthermore, for any value of $M$ and $\gamma$, our rate can guarantee convergence for a wider class of problems. The proof of this theorem, along with justifications for the improved rate, is presented in Appendix H. Mofakhami et al. [2023] derive the

**Results.** The outcomes of this case study are shown in Figure 3. For larger window sizes ($\tau$), we omit the initial iterations in the figure because they follow the same update rule as smaller $\tau$ methods, leading to identical values. Figure 3 demonstrates the advantage of using older snapshots in the optimization process. As the window size increases from 1 to 2, we observe a near-half reduction in the loss shift, particularly in the early iterations, with the improvement persisting even after 50 iterations. While larger windows continue to reduce the loss shift, the marginal gains decrease as window size increases. This is evident from the similarity between the curves for window sizes $t/2$, and 'all'. The decreasing marginal gains elicit a trade-off against the time, memory, and resource consumption. As the window size increases, both time per iteration and the memory consumption increase linearly. Thus, the user has to pick the right aggregation window $\tau$ based on the application and the resources available to achieve the desired convergence speed while respecting the logistical constraints. The corresponding performative risk plot can also be found in Appendix I.

## 8 Conclusion

In this paper, we introduced a new class of algorithms for improved convergence in performative prediction by utilizing historical datasets from previous retraining snapshots. Our theoretical contributions include establishing a new upper bound for last-iterate methods, demonstrating the tightness of this bound, and surpassing existing lower bounds through the aggregation of historical datasets. We have also presented the first lower bound analysis for Repeated Risk Minimization (RRM) within the class of Affine Risk Minimizers. Our empirical results validate the theoretical findings, showing that using prior snapshots leads to more effective convergence to a stable point. These contributions provide new insights into performative prediction and offer an alternative approach to enhancing learning in dynamic environments.

## Acknowledgments

This research was supported by the Canada CIFAR AI Chair Program and the NSERC Discovery Grants RGPIN-2023-04373 and RGPIN-2025-05123. We gratefully acknowledge the computational resources provided by Calcul Quebec (`calculquebec.ca`) and the Digital Research Alliance of Canada (`alliancecan.ca`), which enabled part of the experiments. Simon Lacoste-Julien is a CIFAR Associate Fellow in the Learning in Machines & Brains program. We also thank António Góis and Alan Milligan for their valuable feedback, which notably contributed to this work.

---

[4]Give me Some Credit Dataset, 2011: https://www.kaggle.com/c/GiveMeSomeCredit

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

# A  Auxiliary Lemmas and Technical Results

**Lemma 2** *(Expectation of a Gaussian-Weighted Exponential Function) Let $\mathbf{x} \sim N(\boldsymbol{\mu}, \sigma^2 \boldsymbol{I})$. Then the expected value of $\mathbf{x} \exp\left(-\frac{1}{2e}\|\mathbf{x}\|^2\right)$ is given by:*

$$\mathbb{E}\left[\mathbf{x}\exp\left(-\frac{1}{2e}\|\mathbf{x}\|^2\right)\right] = \exp\left(-\frac{\|\boldsymbol{\mu}\|^2}{2\sigma^2}\left(1 - \frac{1}{\sigma^2\left(\frac{1}{e}+\frac{1}{\sigma^2}\right)}\right)\right) \cdot \frac{\boldsymbol{\mu}}{\sigma^2\left(\frac{1}{e}+\frac{1}{\sigma^2}\right)}.$$

**Proof:** The expected value is expressed as:

$$\mathbb{E}\left[\mathbf{x}\exp\left(-\frac{1}{2e}\|\mathbf{x}\|^2\right)\right] = \int_{\mathbb{R}^n} \mathbf{x}\exp\left(-\frac{1}{2e}\|\mathbf{x}\|^2\right)\frac{1}{(2\pi\sigma^2)^{n/2}}\exp\left(-\frac{1}{2\sigma^2}\|\mathbf{x}-\boldsymbol{\mu}\|^2\right)d\mathbf{x}.$$

Merging the exponentials:

$$\exp\left(-\frac{1}{2}\left(\frac{1}{e}+\frac{1}{\sigma^2}\right)\|\mathbf{x}\|^2 + \frac{1}{\sigma^2}\mathbf{x}^T\boldsymbol{\mu}\right)\cdot\exp\left(-\frac{\|\boldsymbol{\mu}\|^2}{2\sigma^2}\right).$$

Completing the square yields:

$$\exp\left(-\frac{1}{2}\left(\frac{1}{e}+\frac{1}{\sigma^2}\right)\left\|\mathbf{x}-\frac{\boldsymbol{\mu}/\sigma^2}{\frac{1}{e}+\frac{1}{\sigma^2}}\right\|^2\right)\cdot\exp\left(-\frac{\|\boldsymbol{\mu}\|^2}{2\sigma^2}\left(1 - \frac{1}{\sigma^2\left(\frac{1}{e}+\frac{1}{\sigma^2}\right)}\right)\right)$$

Since the integral is over a Gaussian distribution with mean $\frac{\boldsymbol{\mu}/\sigma^2}{\frac{1}{e}+\frac{1}{\sigma^2}}$, after multiplying by the constant term, we obtain:

$$\mathbb{E}[\mathbf{x}\exp\left(-\frac{1}{2e}\|\mathbf{x}\|^2\right)] = \exp\left(-\frac{\|\boldsymbol{\mu}\|^2}{2\sigma^2}\left(1 - \frac{1}{\sigma^2\left(\frac{1}{e}+\frac{1}{\sigma^2}\right)}\right)\right)\cdot\frac{\boldsymbol{\mu}}{\sigma^2\left(\frac{1}{e}+\frac{1}{\sigma^2}\right)}.$$

**Lemma 3** *(Young's Product Inequality) Let $a, b \geq 0$ and let $p, q > 1$ be conjugate exponents, i.e. $\frac{1}{p}+\frac{1}{q}=1$. Then*

$$ab \leq \frac{a^p}{p} + \frac{b^q}{q},$$

*which, as a result, one can derive,*

$$(a+b)^2 \leq 2a^2 + 2b^2.$$

**Lemma 4** *(Bound on Chi-Square Divergence for Convex Combinations) Let $P$ and $Q$ and $R$ be probability distributions on $\mathbb{R}^n$. For any $\alpha \in [0, 1]$, the following inequality holds:*

$$\chi^2\big(\alpha P + (1-\alpha)R, \ \alpha Q + (1-\alpha)R\big) \leq \alpha^3\,\chi^2(P, Q) + 2\,\alpha^2(1-\alpha)\left[\chi^2(P, R) + \chi^2(Q, R)\right].$$

**Proof:** We begin by expanding the chi-square divergence using its definition, followed by applying Young's inequality.

$$\begin{aligned}
\chi^2(\alpha P + (1-\alpha)R, \alpha Q + (1-\alpha)R) &= \int_{-\infty}^{\infty}\frac{(\alpha p(x) + (1-\alpha)r(x) - (\alpha q(x) + (1-\alpha)r(x)))^2}{\alpha q(x) + (1-\alpha)r(x)}\,dx \\
&= \alpha^2\int_{-\infty}^{\infty}\frac{(p(x)-q(x))^2}{\alpha q(x) + (1-\alpha)r(x)}\,dx \\
&\leq \alpha^3\int_{-\infty}^{\infty}\frac{(p(x)-q(x))^2}{q(x)} + \alpha^2(1-\alpha)\int_{-\infty}^{\infty}\frac{(p(x)-q(x))^2}{r(x)}\,dx \\
&\quad \text{(by convexity of }\frac{1}{x}\text{ for positive }x) \\
&\leq \alpha^3\chi^2(P, Q) + 2\alpha^2(1-\alpha)\chi^2(P, R) + 2\alpha^2(1-\alpha)\chi^2(Q, R) \\
&\quad \text{(by Young's inequality Lemma 3 on the last term)}
\end{aligned}$$

**Lemma 5** *(Inverse of an antisymmetric of a Jordan Normal Form Matrix) Let $A \in \mathbb{R}^{d \times d}$ be defined as:*

$$
A = \begin{bmatrix}
1 & 0 & 0 & \ldots & 0 \\
1 & 1 & 0 & \ldots & 0 \\
0 & 1 & 1 & \ldots & 0 \\
\vdots & \vdots & \ddots & \ddots & \vdots \\
0 & \ldots & 0 & 1 & 1
\end{bmatrix},
$$

*and let $bI - cA$ be an invertible matrix where $\frac{c}{b} \leq \frac{1}{2}$ and $A$ is as defined above. Then the inverse of $(bI - cA)$ applied to $e_1$, the first standard basis vector, has the following form for large $d$:*

$$
\mathbf{v} = (bI - cA)^{-1} \frac{e_1}{L} = \frac{1}{cL} \begin{bmatrix}
(\frac{b}{c} - 1)^{-1} \\
(\frac{b}{c} - 1)^{-2} \\
(\frac{b}{c} - 1)^{-3} \\
\vdots \\
(\frac{b}{c} - 1)^{-d}
\end{bmatrix}.
$$

*Moreover, the sum below is:*

$$
\sum_{i=t}^{d} \mathbf{v}_i = \Omega\left(\left(\frac{c}{b}\right)^t\right),
$$

*for $d \geq 2T$ when $T$ is large, and $t \leq T$.*

**Proof:** The matrix $A$ has the following form:

$$
A = \begin{bmatrix}
1 & 0 & 0 & \ldots & 0 \\
1 & 1 & 0 & \ldots & 0 \\
0 & 1 & 1 & \ldots & 0 \\
\vdots & \vdots & \ddots & \ddots & \vdots \\
0 & \ldots & 0 & 1 & 1
\end{bmatrix}.
$$

Thus, $(bI - cA)$ takes the form:

$$
bI - cA = c \begin{bmatrix}
\frac{b}{c} - 1 & 0 & 0 & \ldots & 0 \\
-1 & \frac{b}{c} - 1 & 0 & \ldots & 0 \\
0 & -1 & \frac{b}{c} - 1 & \ldots & 0 \\
\vdots & \vdots & \ddots & \ddots & \vdots \\
0 & \ldots & 0 & -1 & \frac{b}{c} - 1
\end{bmatrix}.
$$

We continue by computing the inverse of the lower triangular matrix with diagonal entries $\lambda_1, \lambda_2, \ldots, \lambda_d$ and subdiagonal entries of $-1$ as shown below:

$$
\begin{bmatrix}
\lambda_1 & 0 & 0 & \ldots & 0 \\
-1 & \lambda_2 & 0 & \ldots & 0 \\
0 & -1 & \lambda_3 & \ldots & 0 \\
\vdots & \vdots & \ddots & \ddots & \vdots \\
0 & \ldots & 0 & -1 & \lambda_d
\end{bmatrix}^{-1}
=
\begin{bmatrix}
\lambda_1^{-1} & 0 & 0 & \ldots & 0 \\
\lambda_1^{-1}\lambda_2^{-1} & \lambda_2^{-1} & 0 & \ldots & 0 \\
\lambda_1^{-1}\lambda_2^{-1}\lambda_3^{-1} & \lambda_2^{-1}\lambda_3^{-1} & \lambda_3^{-1} & \ldots & 0 \\
\vdots & \vdots & \ddots & \ddots & \vdots \\
\lambda_1^{-1}\lambda_2^{-1}\ldots\lambda_d^{-1} & & \ldots & \lambda_{d-1}^{-1}\lambda_d^{-1} & \lambda_d^{-1}
\end{bmatrix}.
$$

Using the formula above (diagonal entries $\lambda_1 = \frac{b}{c} - 1, \lambda_2 = \frac{b}{c} - 1, \ldots, \lambda_d = \frac{b}{c} - 1$ and subdiagonal entries of $-1$) the inverse of $bI - cA$ will have the form:

$$
\frac{1}{c} \begin{bmatrix}
(\frac{b}{c} - 1)^{-1} & 0 & 0 & \ldots & 0 \\
(\frac{b}{c} - 1)^{-2} & (\frac{b}{c} - 1)^{-1} & 0 & \ldots & 0 \\
(\frac{b}{c} - 1)^{-3} & (\frac{b}{c} - 1)^{-2} & (\frac{b}{c} - 1)^{-1} & \ldots & 0 \\
\vdots & \vdots & \ddots & \ddots & \vdots \\
(\frac{b}{c} - 1)^{-d} & & \ldots & (\frac{b}{c} - 1)^{-2} & (\frac{b}{c} - 1)^{-1}
\end{bmatrix}.
$$

Now, applying this inverse to the vector $\frac{e_1}{L}$, where $e_1 = \begin{bmatrix} 1 & 0 & \dots & 0 \end{bmatrix}^T$, we get the following:

$$\mathbf{v} = (bI - cA)^{-1}\frac{e_1}{L} = \frac{1}{cL}\begin{bmatrix} (\frac{b}{c} - 1)^{-1} \\ (\frac{b}{c} - 1)^{-2} \\ (\frac{b}{c} - 1)^{-3} \\ \vdots \\ (\frac{b}{c} - 1)^{-d} \end{bmatrix}.$$

Sum of the entries from index $t$ to $d$ is:

$$\sum_{i=t}^{d} \mathbf{v}_i = \frac{1}{cL}\left( (\frac{b}{c} - 1)^{-t} + (\frac{b}{c} - 1)^{-t-1} + \cdots + (\frac{b}{c} - 1)^{-d} \right).$$

This is a geometric series. The closed form of the sum is:

$$\sum_{i=t}^{d} \mathbf{v}_i = \frac{1}{cL} \cdot (\frac{b}{c} - 1)^{-t} \cdot \frac{1 - (\frac{b}{c} - 1)^{-(d-t+1)}}{1 - (\frac{b}{c} - 1)^{-1}}.$$

For large $d \geq 2t$ and $\frac{b}{c} - 1 \geq 1$, this sum can be approximated by the leading term:

$$\sum_{i=t}^{d} \mathbf{v}_i \approx \frac{1}{cL} \cdot (\frac{b}{c} - 1)^{-t} \cdot \frac{1}{1 - (\frac{b}{c} - 1)^{-1}}.$$

Thus, applying the inequality $\frac{1}{\frac{1}{x} - 1} \leq x$ for all $x < 1$, we obtain the following lower bound for the sum:

$$\sum_{i=t}^{d} \mathbf{v}_i = \Omega\left( \frac{1}{cL} \cdot \left(\frac{c}{b}\right)^t \right).$$

**Lemma 6** *Let $\mathcal{N}(\boldsymbol{\mu}_1, \Sigma_1)$ and $\mathcal{N}(\boldsymbol{\mu}_2, \Sigma_2)$ be two multivariate normal distributions with means $\boldsymbol{\mu}_1, \boldsymbol{\mu}_2 \in \mathbb{R}^d$ and covariance matrices $\Sigma_1, \Sigma_2 \in \mathbb{R}^{d \times d}$. The squared 1-Wasserstein distance between these distributions is bounded by:*

$$W_1^2(\mathcal{N}(\boldsymbol{\mu}_1, \Sigma_1), \mathcal{N}(\boldsymbol{\mu}_2, \Sigma_2)) \leq \|\boldsymbol{\mu}_1 - \boldsymbol{\mu}_2\|^2 + tr\left(\Sigma_1 + \Sigma_2 - 2\left(\Sigma_1\Sigma_2\right)^{1/2}\right).$$

This expression bounds the Wasserstein distance between two multivariate normal distributions, as shown in Dowson and Landau [1982].

**Lemma 7** *Let $\mathcal{N}(\boldsymbol{\mu}_1, \Sigma)$ and $\mathcal{N}(\boldsymbol{\mu}_2, \Sigma)$ be two multivariate normal distributions with means $\boldsymbol{\mu}_1, \boldsymbol{\mu}_2 \in \mathbb{R}^d$ and a shared covariance matrix $\Sigma \in \mathbb{R}^{d \times d}$. The $\chi^2$-divergence between these distributions is bounded by:*

$$\chi^2(\mathcal{N}(\boldsymbol{\mu}_1, \Sigma), \mathcal{N}(\boldsymbol{\mu}_2, \Sigma)) = 1 - e^{-\frac{1}{2}(\boldsymbol{\mu}_1 - \boldsymbol{\mu}_2)^\top \Sigma^{-1}(\boldsymbol{\mu}_1 - \boldsymbol{\mu}_2)} \leq \frac{1}{2}(\boldsymbol{\mu}_1 - \boldsymbol{\mu}_2)^\top \Sigma^{-1}(\boldsymbol{\mu}_1 - \boldsymbol{\mu}_2).$$

This provides the $\chi^2$-divergence between two multivariate normal distributions, as shown in Nielsen and Okamura [2024].

**Lemma 8** *Any projection $proj(.)$ from $\mathbb{R}^d$ into any convex set $\mathcal{C} \in \mathbb{R}^d$ is a continuous function.*

**Proof:** To prove that the projection is continuous, we need to show that if $x_n \to x$ in $\mathbb{R}^d$, then $\text{proj}_\mathcal{C}(x_n) \to \text{proj}_\mathcal{C}(x)$.

Let $y_n = \text{proj}_\mathcal{C}(x_n)$ and $y = \text{proj}_\mathcal{C}(x)$. Since $y_n \in \mathcal{C}$ and $y_n$ minimizes the distance to $x_n$, we have:

$$\|x_n - y_n\| \leq \|x_n - y\| \quad \text{for all } n.$$

As $x_n \to x$, the right-hand side $\|x_n - y\| \to \|x - y\|$, and thus $\|x_n - y_n\|$ is bounded. Since the sequence $\{y_n\}$ is bounded and lies in the compact set $\mathcal{C}$, it has a convergent subsequence $y_{n_k} \to \bar{y} \in \mathcal{C}$. By the continuity of the distance function, we have:

$$\|x - \bar{y}\| = \lim_{k \to \infty} \|x_{n_k} - y_{n_k}\|.$$

As $y = \text{proj}_\mathcal{C}(x)$ minimizes the distance from $x$ to $\mathcal{C}$, it follows that $\bar{y} = y$, and thus $y_n \to y$. Therefore, $\text{proj}_\mathcal{C}(x_n) \to \text{proj}_\mathcal{C}(x)$, proving continuity.

**Lemma 9** *The set of stable points for any method in the class of Affine Risk Minimizers is equivalent to the set of stable points for standard RRM.*

**Proof:** Consider the mapping for an affine risk minimizer using the last $\tau$ iterates, defined as:

$$G_\tau(\theta^{t-1}, \theta^{t-2}, \ldots, \theta^{t-\tau}) = (\theta^t, \theta^{t-1}, \ldots, \theta^{t-\tau+1}),$$

where

$$\theta^t = \arg\min_{\theta \in \Theta} \mathbb{E}_{z \sim D_t}[\ell(f_\theta(x), y)].$$

At a stable point, the mapping satisfies:

$$(\theta^t, \theta^{t-1}, \ldots, \theta^{t-\tau}) = (\theta^t, \theta^{t-1}, \ldots, \theta^{t-\tau+1}),$$

which implies that:

$$\theta^t = \theta^{t-1} = \cdots = \theta^{t-\tau}.$$

We now show that every stable point for this mapping is also a stable point for the standard RRM mapping, defined as:

$$G(\theta^{t-1}) = \theta^t.$$

From the definition of $D_t$, we have:

$$D_t = \sum_{i=t-\tau}^{t-1} \alpha_i^{(t)} D(\theta^i) = D(\theta^{t-1}),$$

since $\sum_{i=t-\tau}^{t-1} \alpha_i^{(t)} = 1$. Therefore:

$$\theta^t = \arg\min_{\theta \in \Theta} \mathbb{E}_{z \sim D_t}[\ell(f_\theta(x), y)] = \arg\min_{\theta \in \Theta} \mathbb{E}_{z \sim D(\theta^{t-1})}[\ell(f_\theta(x), y)] = G(\theta^{t-1}),$$

implying that any stable point for $G_\tau$ is also a stable point for $G$.

Conversely, if $\theta^t = \theta^{t-1}$ at a stable point of $G$, then iterating the mapping $G_\tau$ $\tau$ times yields the sequence:

$$\theta^t = \theta^{t+1} = \cdots = \theta^{t+\tau},$$

A similar argument shows that this stable point satisfies:

$$\theta^{t+\tau} = \arg\min_{\theta \in \Theta} \mathbb{E}_{z \sim D(\theta^{t+\tau-1})}[\ell(f_\theta(x), y)] = \arg\min_{\theta \in \Theta} \mathbb{E}_{z \sim D_{t+\tau}}[\ell(f_\theta(x), y)],$$

because:

$$D_{t+\tau} = \sum_{i=t}^{t+\tau-1} \alpha_i^{(t)} D(\theta^i) = D(\theta^{t-1}).$$

Which leads to,

$$G_\tau(\theta^t, \theta^{t+1}, \ldots, \theta^{t+\tau}) = (\theta^{t-1}, \theta^t, \ldots, \theta^{t+\tau-1})$$

showing that this stable point is also stable for $G_\tau$.

Thus, the set of stable points is equivalent for both mappings.

**Lemma 10** *Let $a$ be a real number with $0 < a < 1$. Then for every integer $t \geq 0$,*

$$a^{\lfloor t/2 \rfloor} \leq a^{-\frac{1}{2}}\left(a^{t/2}\right).$$

**Lemma 11** *Let $A_1$ and $A_2$ be two probability distributions and let $B_1$ and $B_2$ be another two probability distributions. Then, we have*

$$\chi^2\left(\frac{A_1 + A_2}{2}, \frac{B_1 + B_2}{2}\right) \leq \chi^2(A_1, B_1) + \chi^2(A_2, B_2).$$

**Proof:** To compute the $\chi^2$ divergence between the averages $\frac{A_1+A_2}{2}$ and $\frac{B_1+B_2}{2}$, we start with the definition:

$$\chi^2\left(\frac{A_1 + A_2}{2}, \frac{B_1 + B_2}{2}\right) = \int_{-\infty}^{\infty} \frac{\left(\frac{p_1^A(x)+p_2^A(x)}{2} - \frac{p_1^B(x)+p_2^B(x)}{2}\right)^2}{\frac{p_1^B(x)+p_2^B(x)}{2}} \, dx.$$

Simplifying the numerator, we get:

$$= \frac{1}{2} \int_{-\infty}^{\infty} \frac{\left(p_1^A(x) + p_2^A(x) - p_1^B(x) - p_2^B(x)\right)^2}{p_1^B(x) + p_2^B(x)} \, dx.$$

Applying the inequality $(a + b)^2 \leq 2a^2 + 2b^2$, we can further bound this as follows:

$$\leq \frac{1}{2} \int_{-\infty}^{\infty} \frac{2(p_1^A(x) - p_1^B(x))^2 + 2(p_2^A(x) - p_2^B(x))^2}{p_1^B(x) + p_2^B(x)} \, dx.$$

By distributing the terms, this becomes:

$$= \int_{-\infty}^{\infty} \frac{(p_1^A(x) - p_1^B(x))^2}{p_1^B(x) + p_2^B(x)} + \frac{(p_2^A(x) - p_2^B(x))^2}{p_1^B(x) + p_2^B(x)} \, dx.$$

Now, since $\frac{1}{p_1^B(x)+p_2^B(x)} \leq \frac{1}{p_1^B(x)}$ and $\frac{1}{p_1^B(x)+p_2^B(x)} \leq \frac{1}{p_2^B(x)}$, we can split the integral as follows:

$$\leq \int_{-\infty}^{\infty} \frac{(p_1^A(x) - p_1^B(x))^2}{p_1^B(x)} + \frac{(p_2^A(x) - p_2^B(x))^2}{p_2^B(x)} \, dx.$$

By definition of the $\chi^2$ divergence, this final expression is equivalent to:

$$= \chi^2(A_1, B_1) + \chi^2(A_2, B_2).$$

Thus, we have shown that

$$\chi^2\left(\frac{A_1 + A_2}{2}, \frac{B_1 + B_2}{2}\right) \leq \chi^2(A_1, B_1) + \chi^2(A_2, B_2),$$

which completes the proof.

**Lemma 12** *Let $\eta = f_{G(\theta')} - f_{G(\theta)}$. Suppose the function space $\mathcal{F}$ is convex, and*

$$G(\theta) = \arg\min_{\theta' \in \Theta} \mathbb{E}_z\left[\ell(f_{\theta'}(x), y)\right],$$

*where $z = (x, y) \sim p_{f_\theta}$ represents the distribution induced by the model $f_\theta$, and $\ell$ is a continuously differentiable loss function. Then the following inequality holds:*

$$\int \eta(x)^\top \nabla_y \ell(f_{G(\theta)}(x), y) \, p_{f_\theta}(z) \, dz \geq 0.$$

Refer to Mofakhami et al. [2023] for the proof.

**Lemma 13** *Suppose a nonnegative sequence $(a_t)_{t \geq 0}$ satisfies the recurrence*

$$a_{t+1} \leq \epsilon \max\{a_t, a_{t-1}\}$$

*for all $t \geq 1$ and some $0 < \epsilon \leq 1$. Then for every integer $t \geq 0$, one has*

$$a_t \leq \epsilon^{\lfloor t/2 \rfloor} \max\{a_0, a_1\}.$$

**Proof:** Set $A := \max\{a_0, a_1\}$. We proceed by unrolling the recursion in pairs:

$$a_2 \leq \epsilon A, \quad a_3 \leq \epsilon \max\{a_2, a_1\} \leq \epsilon A, \quad a_4 \leq \epsilon \max\{a_3, a_2\} \leq \epsilon^2 A, \ \ldots$$

and in general, each two steps introduce at least one additional factor of $\epsilon$, yielding the claimed bound.

# B Proof of Theorem 1

The proof of Theorem 1 largely follows the approach in Mofakhami et al. [2023]. To facilitate readability, we have restated the common parts from the proof in Mofakhami et al. [2023].

Fix $\theta$ and $\theta'$ in $\Theta$. Let $h : \mathcal{F} \to \mathbb{R}$ and $h' : \mathcal{F} \to \mathbb{R}$ be two functionals defined as follows:

$$h(f_{\hat{\theta}}) = E_{z \sim \mathcal{D}(f_\theta)}[\ell(f_{\hat{\theta}}(x), y)] = \int \ell(f_{\hat{\theta}}(x), y) p_{f_\theta}(z) dz \tag{21}$$

$$h'(f_{\hat{\theta}}) = E_{z \sim \mathcal{D}(f_{\theta'})}[\ell(f_{\hat{\theta}}(x), y)] = \int \ell(f_{\hat{\theta}}(x), y) p_{f_{\theta'}}(z) dz \tag{22}$$

where each data point $z$ is a pair of features $x$ and label $y$.

For a fixed $z = (x, y)$, due to strong convexity of $\ell(f_\theta(x), y)$ in $f_\theta(x)$ we have:

$$\ell(f_{G(\theta)}(x), y) - \ell(f_{G(\theta')}(x), y) \geq \left( f_{G(\theta)}(x) - f_{G(\theta')}(x) \right)^\top \nabla_{\hat{y}} \ell(f_{G(\theta')}(x), y) \\ + \frac{\gamma}{2} \| f_{G(\theta)}(x) - f_{G(\theta')}(x) \|^2. \tag{23}$$

Taking an integral over $z$ w.r.t $p_{f_\theta(z)}$, and knowing that $\| f_{G(\theta)} - f_{G(\theta')} \|_{f_\theta}^2 = \int \| f_{G(\theta)}(x) - f_{G(\theta')}(x) \|^2 p_{f_\theta}(z) dz$, we get the following:

$$h(f_{G(\theta)}) - h(f_{G(\theta')}) \geq \left( \int \left( f_{G(\theta)}(x) - f_{G(\theta')}(x) \right)^\top \nabla_{\hat{y}} \ell(f_{G(\theta')}(x), y) p_{f_\theta}(z) dz \right) \\ + \frac{\gamma}{2} \| f_{G(\theta)} - f_{G(\theta')} \|_{f_\theta}^2. \tag{24}$$

Similarly:

$$h(f_{G(\theta')}) - h(f_{G(\theta)}) \geq \left( \int \left( f_{G(\theta')}(x) - f_{G(\theta)}(x) \right)^\top \nabla_{\hat{y}} \ell(f_{G(\theta)}(x), y) p_{f_\theta}(z) dz \right) \\ + \frac{\gamma}{2} \| f_{G(\theta)} - f_{G(\theta')} \|_{f_\theta}^2. \tag{25}$$

Since $f_{G(\theta)}$ minimizes $h$, the following result can be achieved through the convexity of the function space, (Lemma 12):

$$\int \left( f_{G(\theta')}(x) - f_{G(\theta)}(x) \right)^\top \nabla_{\hat{y}} \ell(f_{G(\theta)}(x), y) p_{f_\theta}(z) dz \geq 0. \tag{26}$$

Adding (24) and (25) and using the above inequality, we conclude:

$$-\gamma \| f_{G(\theta)} - f_{G(\theta')} \|_{f_\theta}^2 \geq \int \left( f_{G(\theta)}(x) - f_{G(\theta')}(x) \right)^\top \nabla_{\hat{y}} \ell(f_{G(\theta')}(x), y) p_{f_\theta}(z) dz. \tag{27}$$

This is a key inequality that will be used later in the proof.

Now recall that there exists $M$ such that $M = \sup_{x,y,\theta} \| \nabla_{\hat{y}} \ell(f_\theta(x), y) \|$ and the distribution map over data is $\epsilon$-sensitive w.r.t Pearson $\chi^2$ divergence, i.e.

$$\chi^2(\mathcal{D}(f_{\theta'}), \mathcal{D}(f_\theta)) \leq \epsilon \| f_\theta - f_{\theta'} \|_{f_\theta}^2. \tag{28}$$

With this in mind, we do the following calculations:

$$\left| \int \left( f_{G(\theta)}(x) - f_{G(\theta')}(x) \right)^\top \nabla_{\hat{y}} \ell(f_{G(\theta')}(x), y) p_{f_\theta}(z) dz - \int \left( f_{G(\theta)}(x) - f_{G(\theta')}(x) \right)^\top \nabla_{\hat{y}} \ell(f_{G(\theta')}(x), y) p_{f_{\theta'}}(z) dz \right|$$

$$= \left| \int \left( f_{G(\theta)}(x) - f_{G(\theta')}(x) \right)^\top \nabla_{\hat{y}} \ell(f_{G(\theta')}(x), y) \left( p_{f_\theta}(z) - p_{f_{\theta'}}(z) \right) dz \right|$$

$$\overset{(*)}{\leq} \int \left| \left( f_{G(\theta)}(x) - f_{G(\theta')}(x) \right)^\top \nabla_{\hat{y}} \ell(f_{G(\theta')}(x), y) \left( p_{f_\theta}(z) - p_{f_{\theta'}}(z) \right) \right| dz$$

$$\leq M \int \left| \| f_{G(\theta)}(x) - f_{G(\theta')}(x) \| \left( p_{f_\theta}(z) - p_{f_{\theta'}}(z) \right) \right| dz$$

$$= M \int \left| \| f_{G(\theta)}(x) - f_{G(\theta')}(x) \| \frac{p_{f_\theta}(z) - p_{f_{\theta'}}(z)}{p_{f_\theta}(z)} p_{f_\theta}(z) \right| dz$$

$$= M \left| \int \| f_{G(\theta)}(x) - f_{G(\theta')}(x) \| \frac{p_{f_\theta}(z) - p_{f_{\theta'}}(z)}{p_{f_\theta}(z)} \left| p_{f_\theta}(z) dz \right| \right.$$

$$\overset{\text{Cauchy-Schwarz Ineq.}}{\leq} M \left( \int \| f_{G(\theta)}(x) - f_{G(\theta')}(x) \|^2 p_{f_\theta}(z) dz \right)^{\frac{1}{2}} \left( \int \left( \frac{p_{f_\theta}(z) - p_{f_{\theta'}}(z)}{p_{f_\theta}(z)} \right)^2 p_{f_\theta}(z) dz \right)^{\frac{1}{2}}$$

$$= M \| f_{G(\theta)} - f_{G(\theta')} \|_{f_\theta} \sqrt{\chi^2(\mathcal{D}(f_{\theta'}), \mathcal{D}(f_\theta))}$$

$(*)$ comes from the fact that $\left| \int f(x) dx \right| \leq \int |f(x)| dx$, and the Cauchy-Schwarz inequality states that $|\mathbb{E}[XY]| \leq \sqrt{\mathbb{E}[X^2]\mathbb{E}[Y^2]}$.
We conclude from the above derivations that:

$$\left| \int \left( f_{G(\theta)}(x) - f_{G(\theta')}(x) \right)^\top \nabla_{\hat{y}} \ell(f_{G(\theta')}(x), y) p_{f_\theta}(z) dz - \int \left( f_{G(\theta)}(x) - f_{G(\theta')}(x) \right)^\top \nabla_{\hat{y}} \ell(f_{G(\theta')}(x), y) p_{f_{\theta'}}(z) dz \right|$$

$$\leq M \| f_{G(\theta)} - f_{G(\theta')} \|_{f_\theta} \sqrt{\chi^2(\mathcal{D}(f_{\theta'}), \mathcal{D}(f_\theta))}. \tag{29}$$

Similar to inequality (26), since $f_{G(\theta')}$ minimizes $h'$, one can prove:

$$\int \left( f_{G(\theta)}(x) - f_{G(\theta')}(x) \right)^\top \nabla_{\hat{y}} \ell(f_{G(\theta')}(x), y) p_{f_{\theta'}}(z) dz \geq 0. \tag{30}$$

From (27) we know that $\int \left( f_{G(\theta)}(x) - f_{G(\theta')}(x) \right)^\top \nabla_{\hat{y}} \ell(f_{G(\theta')}(x), y) p_{f_\theta}(z) dz$ is negative, so with this fact alongside (29) and (30), we can write:

$$\int \left( f_{G(\theta)}(x) - f_{G(\theta')}(x) \right)^\top \nabla_{\hat{y}} \ell(f_{G(\theta')}(x), y) p_{f_\theta}(z) dz \geq -M \| f_{G(\theta)} - f_{G(\theta')} \|_{f_\theta} \sqrt{\chi^2(\mathcal{D}(f_{\theta'}), \mathcal{D}(f_\theta))}. \tag{31}$$

Combining (27) and (31), we obtain:

$$\gamma \| f_{G(\theta)} - f_{G(\theta')} \|_{f_\theta}^2 \leq M \| f_{G(\theta)} - f_{G(\theta')} \|_{f_\theta} \sqrt{\chi^2(\mathcal{D}(f_{\theta'}), \mathcal{D}(f_\theta))}$$

$$\Rightarrow \| f_{G(\theta)} - f_{G(\theta')} \|_{f_\theta} \leq \frac{M}{\gamma} \sqrt{\chi^2(\mathcal{D}(f_{\theta'}), \mathcal{D}(f_\theta))} \overset{(28)}{\leq} \frac{\sqrt{\epsilon} M}{\gamma} \| f_\theta - f_{\theta'} \|_{f_\theta} \tag{32}$$

To prove the existence of a fixed point, we use the Schauder fixed point theorem [Schauder, 1930]. Define

$$\mathcal{U} : f \in \mathcal{F} \to \underset{f' \in \mathcal{F}}{\arg\min} \ \underset{z \sim \mathcal{D}(f)}{\mathbb{E}} \ \ell(f'(x), y).$$

For this function, $\mathcal{U}(f_\theta) = f_{G(\theta)}$. So instead of Equation 32, we can write:

$$\| \mathcal{U}(f_\theta) - \mathcal{U}(f_{\theta'}) \|_{f_\theta} \leq \frac{\sqrt{\epsilon} M}{\gamma} \| f_\theta - f_{\theta'} \|_{f_\theta}. \tag{33}$$

Using Assumption 2, we derive the following bound,

$$\| \mathcal{U}(f_\theta) - \mathcal{U}(f_{\theta'}) \| \leq \left( \sqrt{\frac{C}{c}} \right) \frac{\sqrt{\epsilon} M}{\gamma} \| f_\theta - f_{\theta'} \|. \tag{34}$$

This inequality shows that for any $f_{\theta_0} \in \mathcal{F}$, if $\lim_{n\to\infty} \|f_{\theta_n} - f_{\theta_0}\| = 0$, then $\lim_{n\to\infty} \|\mathcal{U}(f_{\theta_n}) - \mathcal{U}(f_{\theta_0})\| = 0$, which proves the continuity of $\mathcal{U}$ with respect to the norm $\|.\|$. Thus, since $\mathcal{U}$ is a continuous function from the convex and compact set $\mathcal{F}$ to itself, the Schauder fixed point theorem ensures that $\mathcal{U}$ has a fixed point. Therefore, $f_{\theta_{\mathrm{PS}}}$ exists such that $f_{G(\theta_{\mathrm{PS}})} = f_{\theta_{\mathrm{PS}}}$.

If we set $\theta = \theta_{\mathrm{PS}}$ and $\theta' = \theta^{t-1}$ for $\theta_{\mathrm{PS}}$ being any sample in the set of stable classifiers, we know that $G(\theta) = \theta_{\mathrm{PS}}$ and $G(\theta') = \theta^t$. So we will have:

$$\|f_{\theta^t} - f_{\theta_{\mathrm{PS}}}\|_{f_{\theta_{\mathrm{PS}}}} \leq \frac{\sqrt{\epsilon}M}{\gamma}\|f_{\theta^{t-1}} - f_{\theta_{\mathrm{PS}}}\|_{f_{\theta_{\mathrm{PS}}}}. \tag{35}$$

Thus,

$$\|f_{\theta^t} - f_{\theta_{\mathrm{PS}}}\|_{f_{\theta_{\mathrm{PS}}}} \leq \frac{\sqrt{\epsilon}M}{\gamma}\|f_{\theta^{t-1}} - f_{\theta_{\mathrm{PS}}}\|_{f_{\theta_{\mathrm{PS}}}} \leq \left(\frac{\sqrt{\epsilon}M}{\gamma}\right)^t \|f_{\theta_0} - f_{\theta_{\mathrm{PS}}}\|_{f_{\theta_{\mathrm{PS}}}}. \tag{36}$$

Note that Equation 36 applies to any stable point. Suppose there are two distinct stable points, $f_{\theta_{PS}^1}$ and $f_{\theta_{PS}^2}$. By the definition of stable points and using Equation 33, we have:

$$\|\mathcal{U}(f_{\theta_{PS}^1}) - \mathcal{U}(f_{\theta_{PS}^2})\|_{f_{\theta_{PS}^1}} = \|f_{\theta_{PS}^1} - f_{\theta_{PS}^2}\|_{f_{\theta_{PS}^1}} \leq \frac{\sqrt{\epsilon}M}{\gamma}\|f_{\theta_{PS}^1} - f_{\theta_{PS}^2}\|_{f_{\theta_{PS}^1}}.$$

Under the assumption that $\frac{\sqrt{\epsilon}M}{\gamma} < 1$, the inequality above ensures that $f_{\theta_{PS}^1} = f_{\theta_{PS}^2}$[5] and the stable point must be unique. Thus, Equation 36 confirms that RRM converges to a unique stable classifier at a linear rate when $\frac{\sqrt{\epsilon}M}{\gamma} < 1$.

---

[5]It is important to clarify that $f_{\theta_{PS}^1} = f_{\theta_{PS}^2}$ does not imply $\forall x\; f_{\theta_{PS}^1}(x) = f_{\theta_{PS}^2}(x)$. Instead, it indicates that $\|f_{\theta_{PS}^1} - f_{\theta_{PS}^2}\| = 0$.

## C   Proof of Theorem 3

In this section, we examine the tightness of the analysis presented in Perdomo et al. [2020] by considering a specific loss function and designing a particular performativity framework. We focus on the loss function $\ell(z, \theta) = \frac{\gamma}{2}\|\theta - \frac{\beta}{\gamma}z\|^2$, which is $\gamma$-strongly convex with respect to the parameter $\theta$ and its gradient w.r.t. $\theta$ is $\beta$-Lipschitz, aligning with the assumptions stipulated in Perdomo et al. [2020].

We model performativity through the following distribution: $z \sim \mathcal{N}(\epsilon\theta, \sigma^2)$ . According to Lemma 6 the 1-Wasserstein distance between two normal distributions is upper bounded by:

$$W_1(\mathcal{N}(\mu_1, \sigma_1^2), \mathcal{N}(\mu_2, \sigma_2^2)) \leq \sqrt{(\mu_1 - \mu_2)^2} = \epsilon\|\theta_1 - \theta_2\|$$

it follows that the distribution mapping specified is $\epsilon$-sensitive, as described in Perdomo et al. [2020].

Under these conditions, the RRM process results in the following update mechanism:

$$\theta^{t+1} = \epsilon\frac{\beta}{\gamma}\theta^t = (\epsilon\frac{\beta}{\gamma})^t\theta^0$$

This arises because:

$$\theta^{t+1} = \arg\min_{\theta} \mathbb{E}_{z\sim\mathcal{D}(\theta^t)}[\ell(z, \theta)] = \arg\min_{\theta} \mathbb{E}_{z\sim\mathcal{D}(\theta^t)}\left[\frac{\gamma}{2}\theta^2 - \beta\theta z + \frac{\beta^2}{2\gamma}z^2\right]$$

$$= \arg\min_{\theta} \mathbb{E}_{z\sim\mathcal{D}(\theta^t)}\left[\frac{\gamma}{2}\theta^2 - \beta\theta z\right] = \arg\min_{\theta} \frac{\gamma}{2}\theta^2 - \beta\epsilon\theta\theta^t = \epsilon\frac{\beta}{\gamma}\theta^t$$

This progression directly corresponds to the upper bound suggested by Perdomo et al. [2020], confirming that the analysis is tight. No further refinement of the analytical model would mean a faster convergence rate for the given set of assumptions as detailed in Perdomo et al. [2020].

# D  Proof of Theorem 2

We define the model fitting function as $f_\theta(x) = \theta$, and the corresponding loss function is:

$$\ell(x, \theta) = \frac{1}{2\gamma} \left\| \gamma f_\theta(x) - M \operatorname{proj}_{\|.\|=0.95}(x) \right\|^2,$$

where $\operatorname{proj}_{\|.\|=0.95}$ denotes the projection onto the surface of a ball with radius 0.95. By setting $\theta \in \Theta = \{z \mid \|z\| \leq 0.05 \min\{\frac{M}{\gamma}, \frac{1}{\sqrt{\epsilon}}\}\}$, we ensure that the gradient norm remains smaller than $M$. Since the loss function is $\gamma$-strongly convex, it satisfies both Assumptions 4 and 3.

Throughout this proof $\|\theta_1 - \theta_2\| = \|f_{\theta_1} - f_{\theta_2}\|_{f_{\theta'}}$ for any choice of $\theta'$.

We define the distribution mapping as follows:

$$D(\theta) = N\left(\sqrt{\epsilon}\theta, \frac{1}{2}\right),$$

The $\chi^2$-divergence between two distributions $D(\theta_1) = N(\mu_1, \sigma)$ and $D(\theta_2) = N(\mu_2, \sigma)$, where $\mu_1 = \sqrt{\epsilon}\theta_1$ and $\mu_2 = \sqrt{\epsilon}\theta_2$, with $\sigma = \frac{1}{2}$, is given by (Lemma 7):

$$\chi^2(N(\mu_1, \Sigma), N(\mu_2, \Sigma)) \leq \frac{1}{2}(\mu_1 - \mu_2)^\top \Sigma^{-1}(\mu_1 - \mu_2) = \epsilon\|\theta_1 - \theta_2\|^2 = \epsilon\|f_{\theta_1} - f_{\theta_2}\|_{f_{\theta_1}}^2.$$

Thus, the $\chi^2$-divergence between the distributions is bounded by $\epsilon\|\theta_1 - \theta_2\|^2$, making it $\epsilon$-sensitive according to Assumption 1. Note that

With this set up one would derive the update rule:

$$\theta^{t+1} = \operatorname{proj}_\Theta\left(\frac{M}{\gamma}\mathbb{E}\left[\operatorname{proj}(x)\right]\right) = \operatorname{proj}_\Theta\left(\frac{M}{\gamma}\operatorname{erf}\left(\frac{2\mathbb{E}\left[x\right]}{\sqrt{2}}\right)\right)$$

Using,

$$\operatorname{erf}\left(\frac{2x}{\sqrt{2}}\right) \geq x \quad \forall x \leq 0.05,$$

and given that $\mathbb{E}[x] = \sqrt{\epsilon}\theta \leq 0.05\min\left\{\frac{\sqrt{\epsilon}M}{\gamma}, 1\right\}$ by the definition of $\Theta$, the condition holds.

$$\theta^{t+1} \geq \operatorname{proj}_\Theta\left(\frac{M}{\gamma}\mathbb{E}\left[x\right]\right) = \operatorname{proj}_\Theta\left(\frac{M\sqrt{\epsilon}}{\gamma}\theta^t\right)$$

Assuming we start with $\theta^0$ in the feasible set and operate in the regime where $\frac{M\sqrt{\epsilon}}{\gamma} \leq 1$, the projection into the feasible set can be omitted. Therefore, we have:

$$\theta^t \geq \left(\frac{M\sqrt{\epsilon}}{\gamma}\right)^t \theta^0.$$

It is clear that $\theta = 0$ is the stable point in this setup, so:

$$\|\theta^t - \theta_{PS}\| = \Omega\left(\left(\frac{M\sqrt{\epsilon}}{\gamma}\right)^t\right).$$

In other words:

$$\|f_{\theta^t} - f_{\theta_{PS}}\|_{f_{\theta_{PS}}} = \Omega\left(\left(\frac{M\sqrt{\epsilon}}{\gamma}\right)^t\right).$$

For the case where $\frac{M\sqrt{\epsilon}}{\gamma} > 1$, the projection remains constrained to the surface of the ball $\Theta$, preventing convergence to the stable point.

# E    Proof of Lemma 1 and Theorem 4

This proof is heavily inspired by the proof of Theorem 1 in Appendix B. We start by restating the stronger assumption that was added in Lemma 1 and that implies other standard assumptions for this paper.

**Assumption 6** $\epsilon$-*sensitivity with respect to Pearson $\chi^2$ divergence (version 2): The distribution map* $\mathcal{D}(f_\theta)$ *maintains $\epsilon$-sensitivity with respect to Pearson $\chi^2$ divergence. For all $f_\theta, f_{\theta'} \in \mathcal{F}$:*

$$\chi^2(\mathcal{D}(f_{\theta'}), \mathcal{D}(f_\theta)) \leq \frac{\epsilon}{C}\|f_\theta - f_{\theta'}\|^2, \tag{37}$$

*where $\|f_\theta - f_{\theta'}\|^2$ is defined in Equation 8.*

Note that, Assumptions 2 and 6, imply Assumption 1:

$$\chi^2(\mathcal{D}(f_{\theta'}), \mathcal{D}(f_\theta)) \leq \frac{\epsilon}{C}\|f_\theta - f_{\theta'}\|^2 \leq \epsilon\|f_\theta - f_{\theta'}\|_{f_{\theta^*}}^2.$$

Following the methodology described for Theorem 2 in Mofakhami et al. [2023], we begin by defining the functional evaluations at consecutive time steps as follows:

$$h^t(f_{\hat{\theta}}) = \mathbb{E}_{z \sim \mathcal{D}_t}[\ell(f_{\hat{\theta}}, z)] = \int \ell(f_{\hat{\theta}}, z)p_t(z)\, dz,$$

$$h^{t-1}(f_{\hat{\theta}}) = \mathbb{E}_{z \sim \mathcal{D}_{t-1}}[\ell(f_{\hat{\theta}}, z)] = \int \ell(f_{\hat{\theta}}, z)p_{t-1}(z)\, dz,$$

where $p_t(z)$ denotes the probability density function of sample $z$ from the distribution $\mathcal{D}_t$.

Utilizing the convexity of $\ell$ and Lemma 1 from Mofakhami et al. [2023], following the line of argument in Equation 17 of Mofakhami et al. [2023], we establish the following inequality:

$$-\gamma\|f_{\theta^{t+1}} - f_{\theta^t}\|_{p_t}^2 \geq \int (f_{\theta^{t+1}}(x) - f_{\theta^t}(x))^\top \nabla_{\hat{y}}\ell(f_{\theta^t}(x), y)p_t(z)\, dz, \tag{38}$$

where $\|f_{\theta^{t+1}} - f_{\theta^t}\|_{p_t}^2$ represents the squared norm, calculated as:

$$\|f_{\theta^{t+1}} - f_{\theta^t}\|_{p_t}^2 = \int \|f_{\theta^{t+1}}(x) - f_{\theta^t}(x)\|^2 p_t(z)\, dz.$$

and $p_t(x) = \frac{1}{n}\sum_{i=0}^{n-1} p_{f_{\theta^{t-i}}}(x)$, Using the bounded gradient assumption, we deduce:

$$\int (f_{\theta^{t+1}}(x) - f_{\theta^t}(x))^\top \nabla_{\hat{y}}\ell(f_{\theta^t}(x), y)p_t(z)\, dz \geq -M\|f_{\theta^{t+1}} - f_{\theta^t}\|_{p_t}\sqrt{\chi^2(\mathcal{D}_t, \mathcal{D}_{t-1})}. \tag{39}$$

Now combining equations 38 and 39 we get,

$$\gamma\|f_{\theta^{t+1}} - f_{\theta^t}\|_{p_t} \leq M\sqrt{\chi^2(\mathcal{D}_t, \mathcal{D}_{t-1})}. \tag{40}$$

Note that Equation 40, is a direct consequence of Assumptions 3-4, 6, and doesn't rely on definition of $\mathcal{D}_t$ (refer to Equation 32 for the proof). In other words, if we define our method as the mapping

$$\mathcal{U}(f_{\theta_1} \ldots f_{\theta_n}) = \arg\min_{f \in \mathcal{F}} \mathbb{E}_{(x, y) \sim D(f_{\theta_1} \ldots f_{\theta_n})}[\ell(f(x), y)],$$

where $D(f_{\theta_1} \ldots f_{\theta_n}) = \frac{\sum_{i=1}^n D(f_{\theta_i})}{n}$, then,

$$\gamma\|\mathcal{U}(f_{\theta_1} \ldots f_{\theta_n}) - \mathcal{U}(f_{\theta_1'} \ldots f_{\theta_n'})\|_{p_d} \leq M\sqrt{\chi^2(D(f_{\theta_1} \ldots f_{\theta_n}), D(f_{\theta_1'} \ldots f_{\theta_n'}))}, \tag{41}$$

where, $p_d$ is probability density function of distribution $D(f_{\theta_1} \ldots f_{\theta_n})$. We use this information further on in the proof.

The remaining task is to bound the $\chi^2$ divergence. Start by defining:

$$\tilde{D}_t = \frac{1}{n-1}\sum_{i=1}^{n-1} D(f_{\theta^{t-i}})$$

Which implies:

$$D_t = \frac{n-1}{n}\tilde{D}_t + \frac{1}{n}D(f_{\theta^t}) \quad \text{and} \quad D_{t-1} = \frac{1}{n}D(f_{\theta^{t-n}}) + \frac{n-1}{n}\tilde{D}_t$$

Now, using this one can derive:

$$\chi^2(\mathcal{D}_{t-1}, \mathcal{D}_t) \leq (\frac{1}{n})^3 \chi^2(D(f_{\theta^t}), D(f_{\theta^{t-n}}))$$

$$+ 2(\frac{1}{n})^2(\frac{n-1}{n})\left[\chi^2(D(f_{\theta^t}), \tilde{D}_t) + \chi^2(D(f_{\theta^{t-n}}), \tilde{D}_t)\right]$$

(by Lemma 4)

$$\leq (\frac{1}{n})^3 \chi^2(D(f_{\theta^t}), D(f_{\theta^{t-n}}))$$

$$+ 2(\frac{1}{n})^2(\frac{n-1}{n})(\frac{1}{n-1})\sum_{i=1}^{n-1}\chi^2(D(f_{\theta^t}), D(f_{\theta^{t-i}}))$$

$$+ 2(\frac{1}{n})^2(\frac{n-1}{n})(\frac{1}{n-1})\sum_{i=1}^{n-1}\chi^2(D(f_{\theta^{t-n}}), D(f_{\theta^{t-n+i}}))$$

(Proposition 6.1 of Goldfeld et al. [2020], convexity of $f$-divergence w.r.t. its arguments)

$$\leq \frac{\epsilon}{C}(\frac{1}{n})^3 \|f_{\theta^t} - f_{\theta^{t-n}}\|^2$$

$$+ 2(\frac{\epsilon}{C})(\frac{1}{n})^3\sum_{i=1}^{n-1}\|f_{\theta^t} - f_{\theta^{t-i}}\|^2$$

$$+ 2(\frac{\epsilon}{C})(\frac{1}{n})^3\sum_{i=1}^{n-1}\|f_{\theta^{t-n}} - f_{\theta^{t-n+i}}\|^2$$

(by $\epsilon$-sensitivity)

$$\leq m_t^2(\frac{\epsilon}{C})(\frac{1}{n})^2 + 4\,m_t^2(\frac{\epsilon}{C})(\frac{1}{n})^3\sum_{i=1}^{n-1}i^2 = m_t^2(\frac{\epsilon}{C})\left[\frac{1}{n^2} + \frac{2(n-1)(2n-1)}{3n^2}\right]$$

where $m_t^2 = \max_{0\leq i<n}\{\|f_{\theta^{t-i-1}} - f_{\theta^{t-i}}\|^2\}$. Which further gives us,

$$\chi^2(\mathcal{D}_{t-1}, \mathcal{D}_t) \leq \frac{\epsilon m_t^2}{C}\left[\frac{4n^2 - 6n + 5}{3n^2}\right]$$

And in conclusion, we derive the following bound:

$$\|f_{\theta^{t+1}} - f_{\theta^t}\|_{p_t} \leq \frac{\sqrt{\epsilon}Mm_t}{\sqrt{C}\gamma}\left[\frac{4n^2 - 6n + 5}{3n^2}\right]^{\frac{1}{2}}.$$

Using Assumption 2, we further obtain:

$$C\|f_{\theta^{t+1}} - f_{\theta^t}\|_{p_t}^2 := C\int \|f_{\theta^{t+1}}(x) - f_{\theta^t}(x)\|^2 p_t(x)dx$$

$$= \frac{C}{n}\sum_{i=0}^{n-1}\int \|f_{\theta^{t+1}}(x) - f_{\theta^t}(x)\|^2 p_{f_{\theta^{t-i}}}(x)dx \qquad (42)$$

$$= \frac{C}{n}\sum_{i=0}^{n-1}\|f_{\theta^{t+1}} - f_{\theta^t}\|_{f_{\theta^{t-i}}}^2 \geq \|f_{\theta^{t+1}} - f_{\theta^t}\|^2$$

Substituting this back into the previous inequality, we finally get:

$$\|f_{\theta^{t+1}} - f_{\theta^t}\| \leq \frac{\sqrt{\epsilon}Mm_t}{\gamma}\left[\frac{4n^2 - 6n + 5}{3n^2}\right]^{\frac{1}{2}}. \qquad (43)$$

Note the $n = 2$ minimizes the bracket above amongst the integers.[6] Continuing with $n = 2$, we have:

$$\|f_{\theta^{t+1}} - f_{\theta^t}\| \leq \left(\frac{\sqrt{3}}{2}\right) \frac{\sqrt{\epsilon} M m_t}{\gamma}. \tag{44}$$

**Convergence to a Stable Point.** By expanding the $\max$ term in Equation 44, using Lemma 13, we establish the following bound:

$$\|f_{\theta^{t+1}} - f_{\theta^t}\| \leq \left(\frac{\sqrt{3}}{2} \frac{\sqrt{\epsilon} M}{\gamma}\right)^{\lfloor \frac{t}{2} \rfloor} f_0,$$

with $f_0 = \max\{\|f_{\theta^2} - f_{\theta^1}\|, \|f_{\theta^1} - f_{\theta^0}\|\}$. Combining this inequality with Lemma 10 and assuming $\frac{\sqrt{\epsilon} M}{\gamma} \leq 4 \frac{\sqrt{3}}{2}$, we obtain:

$$\|f_{\theta^{t+1}} - f_{\theta^t}\| \leq c \left(\frac{\sqrt{3}}{2} \frac{\sqrt{\epsilon} M}{\gamma}\right)^{\frac{t}{2}} f_0, \tag{45}$$

Where $c = \left(\frac{\sqrt{3}}{2} \frac{\sqrt{\epsilon} M}{\gamma}\right)^{-\frac{1}{2}}$. For clarity, let $\alpha = \left(\frac{\sqrt{3}}{2} \frac{\sqrt{\epsilon} M}{\gamma}\right)^{\frac{1}{2}}$, resulting in:

$$\|f_{\theta^{t+k}} - f_{\theta^t}\| \leq \sum_{i=0}^{k-1} \|f_{\theta^{t+i+1}} - f_{\theta^{t+i}}\| \leq 2\alpha^t \|f_{\theta^1} - f_{\theta^0}\| \left(\sum_{i=0}^{k-1} \alpha^i\right)$$

$$= 2\alpha^t \left(\frac{1 - \alpha^{k-1}}{1 - \alpha}\right) \|f_{\theta^1} - f_{\theta^0}\| \overset{\text{(assuming } \alpha < 1)}{\leq} 2\left(\frac{\alpha^t}{1 - \alpha}\right) \|f_{\theta^1} - f_{\theta^0}\|.$$

Notice that the right-hand side of this inequality is independent of $k$. With $\alpha = \left(\frac{\sqrt{3}}{2} \frac{\sqrt{\epsilon} M}{\gamma}\right)^{\frac{1}{2}} < 1$, for any $\delta > 0$, there exists $t > 1$ such that for all $m > t$, $\|f_{\theta^m} - f_{\theta^t}\| \leq \delta$. Thus, the sequence is Cauchy with respect to the norm $\|\cdot\|$; and by the compactness (and therefore completeness) of $F$, it converges to a point $f^*$.

To show that $f^*$ is a stable point, we start by showing the continuity of the mapping

$$\mathcal{U}(f_{\theta_1}, f_{\theta_2}) = \arg\min_{f \in \mathcal{F}} \mathbb{E}_{(x,y) \sim D(f_{\theta_1}, f_{\theta_2})} \left[\ell(f(x), y)\right],$$

where $D(f_{\theta_1}, f_{\theta_2}) = \frac{D(f_{\theta_1}) + D(f_{\theta_2})}{2}$. Applying Lemma 11 and Assumption 6, we obtain:

$$\chi^2(D(f_{\theta_1}, f_{\theta_2}), D(f_{\theta_1'}, f_{\theta_2'})) \leq \chi^2(D(f_{\theta_1}), D(f_{\theta_1'})) + \chi^2(D(f_{\theta_2}), D(f_{\theta_2'}))$$

$$\leq \frac{\epsilon}{C} \|f_{\theta_1} - f_{\theta_1'}\|^2 + \frac{\epsilon}{C} \|f_{\theta_2} - f_{\theta_2'}\|^2.$$

Combining this with equations 41 and 42, we derive:

$$\gamma^2 \|\mathcal{U}(f_{\theta_1}, f_{\theta_2}) - \mathcal{U}(f_{\theta_1'}, f_{\theta_2'})\|^2 \leq C^2 M^2 \chi^2(D(f_{\theta_1}, f_{\theta_2}), D(f_{\theta_1'}, f_{\theta_2'}))$$

$$\leq \epsilon M^2 (\|f_{\theta_1} - f_{\theta_1'}\|^2 + \|f_{\theta_2} - f_{\theta_2'}\|^2). \tag{46}$$

Thus, for any sequence $\lim_{n \to \infty} (f_{\theta_n^1}, f_{\theta_n^2}) = (f_{\theta^1}, f_{\theta^2})$,

$$\lim_{n \to \infty} \|\mathcal{U}(f_{\theta_n^1}, f_{\theta_n^2}) - \mathcal{U}(f_{\theta^1}, f_{\theta^2})\| \leq \lim_{n \to \infty} \frac{\epsilon M^2}{\gamma^2} (\|f_{\theta_n^1} - f_{\theta^1}\|^2 + \|f_{\theta_n^2} - f_{\theta^2}\|^2) = 0.$$

This implies that if $\lim_{n \to \infty}(f_{\theta_n^1}, f_{\theta_n^2}) = (f_{\theta^1}, f_{\theta^2})$, then $\lim_{n \to \infty} \|\mathcal{U}(f_{\theta_n^1}, f_{\theta_n^2}) - \mathcal{U}(f_{\theta^1}, f_{\theta^2})\| = 0$. By the continuity of $\mathcal{U}$, we conclude:

$$f^* = \lim_{t \to \infty} f_{\theta^{t+1}} = \lim_{t \to \infty} \mathcal{U}(f_{\theta^t}, f_{\theta^{t-1}}) = \mathcal{U}\left(\lim_{t \to \infty} f_{\theta^t}, \lim_{t \to \infty} f_{\theta^{t-1}}\right) = \mathcal{U}(f^*, f^*).$$

This establishes that $f^* = f_{\theta_{PS}}$ is a stable point.

---

[6] We can derive a convergence guarantee for any $n$. The values of $n \leq 5$ yields a bracket of $\leq 1$ and thus can match or improve the class of convergence compared to just the last iterate. For larger $n$'s, we still have convergence, but on a smaller class of functions as the bracket $> 1$.

# F  Proof of Lower bound in Perdomo et al. [2020] Framework

In this proof, we begin by considering a loss function defined as follows:

$$\ell(z,\theta) = \frac{\gamma}{2}\|\theta - \frac{\beta}{\gamma}z\|^2. \tag{47}$$

This function is $\gamma$-strongly convex for the parameter $\theta$ and its gradient with respect to $\theta$ is $\beta$-Lipschitz in sample space. The necessary assumptions on the loss function, as outlined in Perdomo et al. [2020], are satisfied by this formulation.

We define the matrix $A$ within $\mathbb{R}^{d \times d}$ as:

$$A = \begin{bmatrix} 1 & 0 & 0 & \dots & 0 \\ 1 & 1 & 0 & \dots & 0 \\ 0 & 1 & 1 & \dots & 0 \\ \vdots & \vdots & \ddots & \ddots & \vdots \\ 0 & \dots & 0 & 1 & 1 \end{bmatrix}.$$

The critical property of this matrix is that if a vector $b$ The key property of this matrix is that if a vector $b \in \text{span}\{e_i \mid i \leq t\}$, then $Ab \in \text{span}\{e_i \mid i \leq t+1\}$, where each $e_i \in \mathbb{R}^d$ is a standard basis vector with all coordinates zero except for the $i$-th coordinate, which is 1. This structure enables the introduction of a new dimension only at the end of each RRM iteration. With the correct initialization, this ensures that the updates remain within a minimum distance from the stable point due to undiscovered dimensions.

We define $\mathcal{D}(\theta)$ as the distribution of $z$ given by:

$$z \sim \mathcal{N}\left(\frac{\epsilon}{2}A\theta + e_1, \sigma^2\right).$$

Note that since spectral radius $A$ is 2, the mapping $\mathcal{D}(.)$ defined as above would be $\epsilon$-sensitive. Under this setting, the first-order Repeated Risk Minimization (RRM) update, starting with $\theta_0 = e_1$, is described by:

$$\theta^{t+1} = \frac{\beta}{\gamma}\left(\frac{\epsilon}{2}A\theta^t + e_1\right),$$

Due to the properties of matrix $A$, we conclude that $\theta^{t+1} \in \text{span}\{e_i \mid \forall i \leq t+1\}$.

The stationary point $\theta_{PS}$ of this setup is located at:

$$\theta_{PS} = \left(\frac{\gamma}{\beta}I - \frac{\epsilon}{2}A\right)^{-1}e_1,$$

Note that at time step $t$ the best model within the feasible set is $\theta^t \in \text{span}\{e_i \mid \forall i \leq t\}$. Given that one can conclude that the best L1-distance to stationary point achievable at time step $t$ is lower bounded by the sum over the last $d - t$ entries of $\theta_{PS}$. Setting $d = 2T$ and using Lemma 5 we get

$$\|\theta^t - \theta_{PS}\| = \Omega\left((\frac{\epsilon\beta}{2\gamma})^t\right).$$

Similar to Repeated Risk Minimization (RRM), the Repeated Gradient Descent (RGD) method introduces a new dimension in each iteration step. Specifically, the gradient update rule in RGD is given by:

$$\mathbb{E}_{z\sim\mathcal{D}(\theta^t)}\nabla_\theta\ell(z,\theta) = \gamma\theta^t + \beta\left(\frac{\epsilon}{2}A\theta^t + e_1\right),$$

This formulation ensures that each step effectively augments the dimensionality of the parameter space being explored only by a single dimension. Consequently, the lower bound established for RRM also applies to these RGD settings.

## F.1 Lower Bound for Proximal RRM

We proceed to use the exact same setup as Appendix F. Keeping in mind that the only changing factor here would be the optimization oracle at each step of RRM. Consider the general case of

$$\theta^{t+1} = \arg\min_{\theta \in \Theta} \mathbb{E}_{(x,\,y) \sim D_t} \left[ \ell(f_\theta(x),\, y) \right] + \frac{\lambda}{2} \|\theta - \theta^t\|^2 + \frac{\omega}{2} \|\theta\|^2 \tag{48}$$

with $D_t$ defined as:

$$D_t = \sum_{i=0}^{t-1} \alpha_i^{(t)} D(f_{\theta^i}), \quad \text{s.t.} \quad \sum_{i=0}^{t-1} \alpha_i^{(t)} = 1. \tag{49}$$

Now given that the loss is strongly convex, the minimizer retrieves a unique point according to the mapping:

$$\theta^{t+1} = \frac{1}{\gamma + \lambda + \omega} \left( \beta \mathbb{E}_{(x,\,y) \sim D_t}[z] + \lambda \theta^t \right) \tag{50}$$

Same as before, using Lemma 9, we search for the stable point using the standard RRM update rule, which in this setup would be the following:

$$\theta^{t+1} = \frac{1}{\gamma + \lambda + \omega} \left( \beta \left( \frac{\epsilon}{2} A \theta^t + e_1 \right) + \lambda \theta^t \right)$$

This update rule for $\lambda \neq \infty$ would retrieve the stationary point $\theta_{PS}$,

$$\theta_{PS} = \left( \frac{\gamma + \omega}{\beta} I - \frac{\epsilon}{2} A \right)^{-1} e_1,$$

Note that again, because of the structure of matrix $A$, Equation 50 can introduce a new dimension only after taking one step of RRM. Therefore, the best method any method in the ARM class can do is to only match the first $t$ coordinates of the stationary point. So one can conclude that the best L1-distance to stationary point achievable at time step $t$ is lower bounded by the sum over the last $d - t$ entries of $\theta_{PS}$. Setting $d = 2T$ and using Lemma 5 we get

$$\|\theta^t - \theta_{PS}\| = \Omega \left( \left(\frac{\epsilon \beta}{2(\gamma + \omega)}\right)^t \right).$$

# G Proof of Lower Bound for Modified Mofakhami et al. [2023] Framework

We define the model fitting function as $f_\theta(x) = \theta$, and the corresponding loss function is:

$$\ell(x, \theta) = \frac{1}{2\gamma} \left\| \gamma f_\theta(x) - M(1-\delta)xe^{-\frac{1}{2e}\|x\|^2} \right\|^2.$$

This loss is $\gamma$-strongly convex, ensuring unique minimizers and stable convergence properties. Additionally, we assume $\theta \in \Theta = \{z|\|z\| \leq \frac{\delta M}{\gamma}\}$, ensuring that the gradient norm $\|\gamma\theta - M(1-\delta)xe^{-\frac{1}{2e}\|x\|^2}\|$ remains bounded by $M$. This holds because the mapping $f(x) = xe^{-\frac{1}{2e}\|x\|^2}$ is chosen such that, $f : \mathbb{R} \to [0, 1]$.

Observe that, for all $f_{\theta*}, f_\theta, f_{\theta'} \in \mathcal{F}$, we have $\|\theta - \theta'\| = \|f_\theta - f_{\theta'}\|_{f_{\theta*}}$:

$$\|f_\theta - f_{\theta'}\|_{f_{\theta*}}^2 = \int \|f_\theta(x) - f_{\theta'}(x)\|^2 p_{f_{\theta*}}(x)\, dx = \int \|\theta - \theta'\|^2 p_{f_{\theta*}}(x)\, dx = \|\theta - \theta'\|^2.$$

We define the distribution mapping as follows:

$$D(\theta) = N\left(\sqrt{\frac{\sigma^2\epsilon}{2}}A\theta + \frac{e_1}{L}, \sigma^2 I\right),$$

where $A$ is a lower triangular matrix:

$$A = \begin{bmatrix} 1 & 0 & 0 & \dots & 0 \\ 1 & 1 & 0 & \dots & 0 \\ 0 & 1 & 1 & \dots & 0 \\ \vdots & \vdots & \ddots & \ddots & \vdots \\ 0 & \dots & 0 & 1 & 1 \end{bmatrix}.$$

Matrix $A$ has the property that if $b$ is in the span of $\{e_1, \dots, e_i\}$, then $Ab$ will be in the span of $\{e_1, \dots, e_{i+1}\}$. Here, $e_i$ denotes the standard basis vector, where its $i$-th element is 1 and all other elements are 0. This makes $A$ crucial for ensuring that each update step involves interactions that span progressively larger subspaces.

The $\chi^2$-divergence between two distributions $D(\theta_1) = N(\mu_1, \Sigma)$ and $D(\theta_2) = N(\mu_2, \Sigma)$, where $\mu_1 = \sqrt{\frac{\sigma^2\epsilon}{2}}A\theta + \frac{e_1}{L}$ and $\mu_2 = \sqrt{\frac{\sigma^2\epsilon}{2}}A\theta' + \frac{e_1}{L}$, with $\Sigma = \sigma^2 I$, according to Lemma 7:

$$\chi^2(N(\mu_1, \Sigma), N(\mu_2, \Sigma)) \leq \frac{1}{2}(\mu_1 - \mu_2)^\top \Sigma^{-1}(\mu_1 - \mu_2) = \frac{1}{\sigma^2}\left(\sqrt{\frac{\sigma^2\epsilon}{2}}A\right)^2\|\theta_1 - \theta_2\|^2.$$

Since the spectral norm of matrix $A$ is 2, we have:

$$\chi^2(D(\theta_1), D(\theta_2)) \leq \epsilon\|\theta_1 - \theta_2\|^2 = \epsilon\|f_\theta - f_{\theta'}\|_{f_\theta}.$$

Thus, the $\chi^2$-divergence between the distributions is bounded by $\epsilon\|\theta - \theta'\|^2$, ensuring that the divergence scales with the difference between $\theta$ and $\theta'$.

The update rule for $\theta$ is:

$$\theta^{t+1} = \text{proj}_\Theta\left(\frac{M}{\gamma}(1-\delta)\mathbb{E}\left[xe^{-\frac{1}{2e}\|x\|^2}\right]\right)$$

$$= \text{proj}_\Theta\left(\frac{M}{\gamma}(1-\delta)\exp\left(-\frac{\|\mathbb{E}[x]\|^2}{2\sigma^2}\left(1 - \frac{1}{\frac{\sigma^2}{e}+1}\right)\right) \cdot \frac{\mathbb{E}[x]}{\frac{\sigma^2}{e}+1}\right).$$

This is the unique minimizer of the loss function due to the $\gamma$-strong convexity. Additionally, this is a continuous mapping from a compact convex set $\Theta$ to itself. By Schauder's fixed-point theorem, there exists a stable fixed point, denoted as $\theta_{PS}$, satisfying:

$$\theta_{PS} = \text{proj}_\Theta\left(\frac{Mc_{\sigma^2,\theta_{PS}}}{\gamma}(1-\delta)\mathbb{E}[x]\right) = \text{proj}_\Theta\left(\frac{Mc_{\sigma^2,\theta_{PS}}}{\gamma}(1-\delta)\left(\sqrt{\frac{\sigma^2\epsilon}{2}}A\theta_{PS} + \frac{e_1}{L}\right)\right),$$

where $c_{\sigma^2, \theta_{PS}} = \dfrac{\exp\left(-\frac{\|\mathbb{E}[x]\|^2}{2\sigma^2}\left(1 - \frac{1}{\frac{\sigma^2}{e}+1}\right)\right)}{\frac{\sigma^2}{e}+1} \leq 1$. Assuming $\frac{M\sqrt{\epsilon}}{\gamma} \leq 1$ and $\sigma \leq \frac{\sqrt{2}}{2}$ we get that,

$$\left\| \frac{Mc_{\sigma^2, \theta_{PS}}}{\gamma}(1-\delta)\left(\sqrt{\frac{\sigma^2\epsilon}{2}}A\theta_{PS} + \frac{e_1}{L}\right)\right\| \leq \left\| \frac{Mc_{\sigma^2, \theta_{PS}}}{\gamma}(1-\delta)\sqrt{\frac{\sigma^2\epsilon}{2}}A\theta_{PS}\right\| + \frac{Mc_{\sigma^2, \theta_{PS}}(1-\delta)}{\gamma L}$$

$$\leq \left\| \frac{1}{2}(1-\delta)\theta_{PS}\right\| + \frac{M(1-\delta)}{\gamma L}$$

$$\leq \frac{1}{2}(1-\delta)\delta + \frac{M(1-\delta)}{\gamma L}$$

Choosing $L \geq \frac{2M(1-\delta)}{\gamma(\delta+\delta^2)}$ one can guarantee the term in the projection operation would have a norm smaller than $\delta$, i.e. it would be in $\Theta$. So you can drop the projection operation from the equality above.

Thus, the stable point would hold true in the following equality:

$$\theta_{PS} = \left(I - (1-\delta)\frac{c_{\sigma^2, \theta_{PS}}}{\sqrt{2}}\frac{\sqrt{\sigma^2\epsilon}M}{\gamma}A\right)^{-1}\frac{e_1}{L}.$$

The same assumptions stated above would allow us to use Lemma 5:

$$\|\theta^t - \theta_{PS}\| = \Omega\left(\left((1-\delta)\frac{c_{\sigma^2, \theta_{PS}}}{\sqrt{2}}\frac{\sqrt{\sigma^2\epsilon}M}{\gamma}\right)^t\right).$$

To lower bound $c_{\sigma^2, \theta_{PS}}$, we note that $\|\mathbb{E}[x]\| \in [0, \frac{\epsilon}{2}\delta + \frac{\gamma(\delta+\delta^2)}{2M(1-\delta)}]$ and minimize the exponential term with respect to $\sigma^2$:

$$\exp\left(-\frac{\|\mathbb{E}[x]\|^2}{2\sigma^2}\left(1 - \frac{1}{\frac{\sigma^2}{e}+1}\right)\right) \geq \exp\left(-c\delta\right),$$

Where $c > 0$ is a constant independent of $\delta$. Setting $\sigma = \frac{\sqrt{2}}{2}$ to maximise $\frac{\sigma}{\frac{\sigma^2}{e}+1}$, and $\lim \delta \to 0$, we achieve:

$$\|\theta^t - \theta_{PS}\| = \Omega\left(\left(\frac{1}{\frac{1}{e}+2}\frac{\sqrt{\epsilon}M}{\gamma}\right)^t\right).$$

Hence,

$$\|f_{\theta^t} - f_{\theta_{PS}}\|_{f_{\theta_{PS}}} = \Omega\left(\left(\frac{1}{\frac{1}{e}+2}\frac{\sqrt{\epsilon}M}{\gamma}\right)^t\right).$$

# H  Proof of Theorem 8

Consider a feature vector $x$ divided into strategic features $x_s$ and non-strategic features $x_f$, so that $x = (x_s, x_f)$. We resample only the strategic features with probability $g(f_\theta(x))$, representing the probability of rejection for $x$. The pdf of the modified distribution $p_{f_\theta}$ is:

$$p_{f_\theta}(x) = p(x)\,(1 - g(f_\theta(x))) + \int_{x'_s} p(x'_s, x_f)\, g(f_\theta(x'_s, x_f))\, p(x_s)dx'_s,$$

where the integral is over all possible values of $x'_s$ with $x_f$ held constant, since only the strategic features are resampled. The first term represents the option that we accept the first sample at $x$; the second term represents the possibility that we reject the first sample at $x' = (x'_s, x_f)$ and then resample at $x_s$ to obtain $x$ as well.

Assuming the strategic and non-strategic features are independent, we can rewrite this expression as:

$$
\begin{aligned}
p_{f_\theta}(x) =& p(x)(1 - g(f_\theta(x))) + \int_{x'_s} p(x'_s, x'_f = x_f)\, g(f_\theta(x'))\, p(x_s)dx'_s \\
=& p(x)(1 - g(f_\theta(x))) + \int_{x'_s} g(f_\theta(x'))p_{X_s}(x'_s)p_{X_f}(x_f)p_{X_s}(x_s)dx'_s \\
=& p(x)(1 - g(f_\theta(x))) + \int_{x'_s} g(f_\theta(x'))p_{X_s}(x'_s)p(x)dx'_s \\
=& p(x)\left((1 - g(f_\theta(x))) + \int_{x'_s} g(f_\theta(x'))p_{X_s}(x'_s)dx'_s\right) \\
=& p(x)\,(1 - g(f_\theta(x)) + C_\theta(x_f)),
\end{aligned}
\tag{51}
$$

where $p_{X_s}$ and $p_{X_f}$ are the marginal distributions of the strategic and non-strategic features, respectively, and we define:

$$C_\theta(x_f) = \int_{x'_s} p_{X_s}(x'_s)\, g(f_\theta(x'_s, x_f))\, dx'_s.$$

Since $0 \le f_\theta(x) \le 1 - \delta$ for some $\delta > 0$, it follows that $\delta \le g(f_\theta(x)) \le 1$ for every $x$. Therefore, $\delta \le C_\theta(x_f) \le 1$.

In the RIR procedure, the distribution of the label $y$ given $x$ is not affected by the predictions so for every $z = (x, y)$ we have $p_{f_\theta}(z) = p_{f_\theta}(x)p(y|x)$ for any $f_\theta$. This results in the following equality:

$$\chi^2(\mathcal{D}(f_{\theta'}), \mathcal{D}(f_\theta)) = \int \frac{(p_{f_{\theta'}}(z) - p_{f_\theta}(z))^2}{p_{f_\theta}(z)}dz = \int \frac{(p_{f_{\theta'}}(x) - p_{f_\theta}(x))^2}{p_{f_\theta}(x)}dx$$

We prove that this mapping is $\epsilon$-sensitive with respect to $\chi^2$ divergence, where $\epsilon = \dfrac{1}{\delta}\left(1 + \dfrac{1 - \delta}{2\sqrt{\delta}}\right)$.

$$
\begin{aligned}
\chi^2(\mathcal{D}(f_{\theta'}), \mathcal{D}(f_\theta)) =& \int \frac{\left(p_{f_{\theta'}}(x) - p_{f_\theta}(x)\right)^2}{p_{f_\theta}(x)}dx \\
=& \int \frac{p(x)^2\left[f_\theta(x) - f_{\theta'}(x) - (C_\theta(x_f) - C_{\theta'}(x_f))\right]^2}{p(x)\left(1 - f_\theta(x) - \delta + C_\theta(x_f)\right)}dx \\
\le& \frac{1}{\delta}\int p(x)\Big[\left(f_\theta(x) - f_{\theta'}(x)\right)^2 + (C_\theta(x_f) - C_{\theta'}(x_f))^2 \\
& \qquad\qquad - 2\left(f_\theta(x) - f_{\theta'}(x)\right)(C_\theta(x_f) - C_{\theta'}(x_f))\Big]dx
\end{aligned}
$$

This inequality follows from the fact that $\delta \le C_\theta(x_f)$ and $1 - g(f_\theta(x)) \ge 0$, therefore $\frac{1}{1 - g(f_\theta(x)) + C_\theta(x_f)} \le \frac{1}{\delta}$.

Continuing, we have:

$$= \frac{1}{\delta} \left[ \int p(x) \left( f_\theta(x) - f_{\theta'}(x) \right)^2 dx \right.$$

$$+ \int_{x_f} p_{X_f}(x_f) \left( C_\theta(x_f) - C_{\theta'}(x_f) \right)^2 dx_f$$

$$\left. - 2 \int_{x_f} p_{X_f}(x_f) \left( C_\theta(x_f) - C_{\theta'}(x_f) \right) \int_{x_s} p_{X_s}(x_s) \left( f_\theta(x) - f_{\theta'}(x) \right) dx_s \, dx_f \right]$$

$$= \frac{1}{\delta} \left[ \int p(x) \left( f_\theta(x) - f_{\theta'}(x) \right)^2 dx - \int_{x_f} p_{X_f}(x_f) \left( C_\theta(x_f) - C_{\theta'}(x_f) \right)^2 dx_f \right]$$

$$\leq \frac{1}{\delta} \int p(x) \left( f_\theta(x) - f_{\theta'}(x) \right)^2 dx$$

This comes from the fact that $\int_{x'_s} p_{X_s}(x'_s)(f_\theta((x'_s, x_f)) - f_{\theta'}((x'_s, x_f)))dx'_s = C_\theta(x_f) - C_{\theta'}(x_f)$. We use equation 51 to replace $p(x)$.

$$= \frac{1}{\delta} \int \left( p_{f_\theta}(x) + p(x) \left( f_\theta(x) + \delta - C_\theta(x_f) \right) \right) \left( f_\theta(x) - f_{\theta'}(x) \right)^2 dx$$

$$= \frac{1}{\delta} \| f_\theta - f_{\theta'} \|^2_{f_\theta} + \frac{1}{\delta} \int p(x) \left( f_\theta(x) + \delta - C_\theta(x_f) \right) \left( f_\theta(x) - f_{\theta'}(x) \right)^2 dx$$

$$\overset{\text{Cauchy-Schwarz}}{\leq} \frac{1}{\delta} \| f_\theta - f_{\theta'} \|^2_{f_\theta} + \frac{1}{\delta} \left( \int p(x) \left( f_\theta(x) + \delta - C_\theta(x_f) \right)^2 dx \right)^{1/2} \left( \int p(x) \left( f_\theta(x) - f_{\theta'}(x) \right)^4 dx \right)^{1/2}$$

$$\leq \frac{1}{\delta} \| f_\theta - f_{\theta'} \|^2_{f_\theta} + \frac{1}{\delta} \left( \int_{x_f} p_{X_f}(x_f) Var_{x_s} \left[ g(f_\theta(x)) \right] dx_f \right)^{1/2} \left( \int p(x) \left( f_\theta(x) - f_{\theta'}(x) \right)^4 dx \right)^{1/2}$$

Since $g(f_\theta(x))$ is a bounded random variable in $[\delta, 1]$, its variance is less than $\frac{(1-\delta)^2}{4}$, according to Popoviciu's inequality. Also since for any $\theta \in \Theta$ we have $f_\theta(x) \leq 1$ we can infer $|f_\theta(x) - f_{\theta'}(x)| \leq 1$

$$\leq \frac{1}{\delta} \| f_\theta - f_{\theta'} \|^2_{f_\theta} + \frac{1-\delta}{2\delta} \left( \int p(x) \left( f_\theta(x) - f_{\theta'}(x) \right)^4 dx \right)^{1/2}$$

$$\leq \frac{1}{\delta} \| f_\theta - f_{\theta'} \|^2_{f_\theta} + \frac{1-\delta}{2\delta} \left( \int p(x) \left( f_\theta(x) - f_{\theta'}(x) \right)^2 dx \right)^{1/2}$$

$$\leq \frac{1}{\delta} \| f_\theta - f_{\theta'} \|^2_{f_\theta} + \frac{1-\delta}{2\delta} \| f_\theta - f_{\theta'} \|$$

From Appendix A.3 in Mofakhami et al. [2023], we know that $\| f_\theta - f_{\theta'} \|^2 \leq \frac{1}{\delta} \| f_\theta - f_{\theta'} \|^2_{f_\theta}$. Hence,

$$\chi^2(\mathcal{D}(f_{\theta'}), \mathcal{D}(f_\theta)) \leq \frac{1}{\delta} \left( 1 + \frac{1-\delta}{2\sqrt{\delta}} \right) \| f_\theta - f_{\theta'} \|^2_{f_\theta}$$

**Rate Improvement Arguments:** By using Assumptions 4 and 6 from Mofakhami et al. [2023], it can be shown that the method is $C\epsilon$-sensitive as defined in Assumption 1. Specifically,

$$\chi^2(\mathcal{D}(f_{\theta'}), \mathcal{D}(f_\theta)) \leq \epsilon \| f_\theta - f_{\theta'} \|^2 \leq C\epsilon \| f_\theta - f_{\theta'} \|^2_{f_\theta}.$$

In this case, our rate aligns with the rate from Mofakhami et al. [2023], demonstrating that in all cases where their rate holds, our approach offers at least an equivalent or faster rate. However, there are instances where our rate results in a smaller constant than $C\epsilon$. As outlined in Appendix A.3 of Mofakhami et al. [2023], the same RIR framework derives $C = \frac{1}{\delta}$ under Assumption 2 and $\epsilon = \frac{1}{\delta}$ with respect to Assumption 6, yielding $C\epsilon = \frac{1}{\delta^2}$. We show that instead of $C\epsilon = \frac{1}{\delta^2}$, we obtain $\frac{1}{\delta} \left( 1 + \frac{1-\delta}{2\sqrt{\delta}} \right)$, which is strictly smaller for any $0 \leq \delta < 1$. This shows that this rate is a strict improvement over Mofakhami et al. [2023].

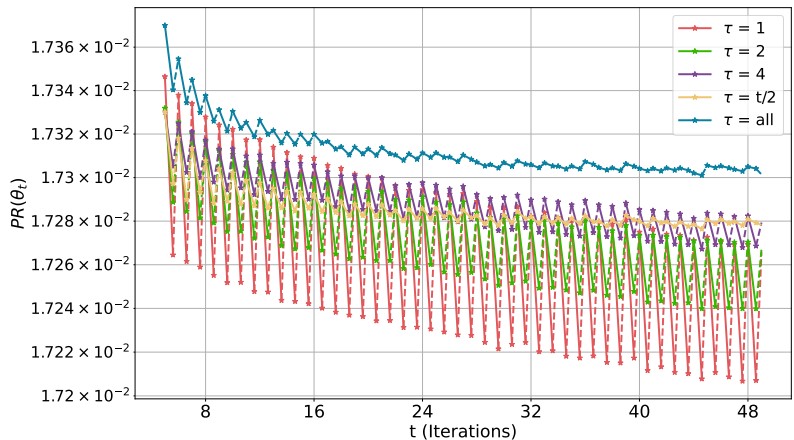

Figure 4: Log performative risk for the credit scoring environment across the RRM iterations. The numbers in the plot are averaged over 500 runs. **Increasing the size of aggregation window $\tau$ from $1 \to 2 \to 4 \to t/2 \to all$ reduces the oscillations in the risk and converges to the same point.** Note that the plot starts from iteration 5 for better readability as the initial risk values were very high.

## I   Performative Risk for Credit-Scoring

Figure 4 shows the log performative risk for the credit-scoring environment. This metric has been adapted from Mofakhami et al. [2023]. Figure 4 further substantiates our claims as we see lower oscillations in the risk for larger aggregation windows. We also note that although larger windows yield more stable trajectories, they can incur worse performative risk along the way before ultimately settling. Furthermore, methods converge to roughly the same performatively stable point—as predicted by the theory of a unique stable point in Lemma 9—since the difference in log performative risk at the end of 50 iterations is negligible between all methods. However, as pointed out in Section 7, all methods oscillate in a similar range, thus hindering the readability of the plot.

**Hyper-parameters.** For our experiments, we fix the value of $\delta = 0.55$. The RRM procedure is carried out for a maximum of 50 iterations with a learning rate of $3e\text{-}4$ and Adam optimizer run over A100-40G GPUs. Each run only took a few minutes. Further, all the experimental results and plots are averaged over 500 runs, where each run for each method has the same model initialization. Thus, the only source of randomization is the sampling under *RIR* mechanism, where the sampling changes across different runs but is the same for all the methods given a specific run.

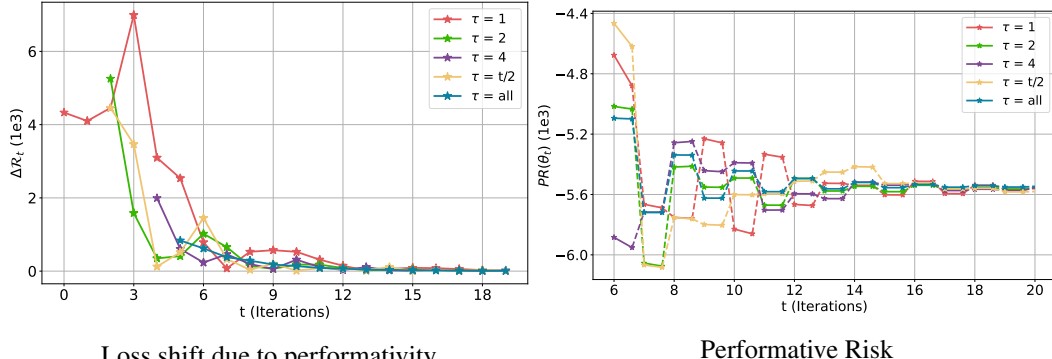

Loss shift due to performativity       Performative Risk

Figure 5: The plot shows loss shift due to performativity and performative risk across the iterations for player 1 in the game between two firms. The values in the plot are means over 200 runs. Increasing the aggregation window size $\tau$ leads to lower loss shifts even in this simple game and hence, faster convergence than just relying on the dataset from the current timestamp.

## J    Revenue Maximization in Ride Share Market

**Setup.** This is a two-player semi-synthetic game between two ride-share providers, Uber and Lyft, both trying to maximize their respective revenues. Each player takes an action in this game by setting their price for the riders across 11 different locations in the same city of Boston, MA. The price set by one firm directly influences the demand observed by both firms. The demand constitutes the data distribution and at each time step, a total of 25 demand samples are sampled for a firm $i$, and the optimal response is found by minimizing Equation 53 for a maximum of 40 retraining steps on CPU. Each run took only a few seconds. The simulations use the publicly available *Uber and Lyft dataset from Boston, MA* on Kaggle[7].

**Notations and Equations.** Let $i = 1, 2$ denote the two firms in the game. Inspired by Narang et al. [2024], each firm $i$ observes a demand $z_i$ that depends linearly on the firm's price $\mathbf{x_i}$ and its opponent's price $\mathbf{x_{-i}}$ as follows:

$$\mathbf{z}_i = \mathbf{A}_i \mathbf{x}_i + \mathbf{A}_{-i} \mathbf{x}_{-i} + \xi, \quad \xi \sim \mathcal{N}(\mathbf{z}_{\text{base}}, 1) \tag{52}$$

where $\mathbf{z_{base}}$ is the mean demand observed at each of the 11 locations, as measured in the kaggle dataset. Each demand sample is a vector of dimension 11.
$\mathbf{A_i}$ and $\mathbf{A_{-i}}$ are fixed matrices representing the price elasticity of demand, i.e. the change in demand due to a unit change in price for the player $i$ and $-i$ (opponent) respectively. We introduce interactions between the ride prices in a location and the demand in a different location within the same city by making $\mathbf{A}$ matrices non-diagonal. Additionally, note that the price elasticities $\mathbf{A_i}$ will always be negative as the firm will experience less demand if it increases its price. Similarly, the price elasticities $\mathbf{A_{-i}}$ will be positive.
Each player observes a revenue of $\mathbf{z_i^T x_i}$. Thus, the loss function that each player $i$ seeks to minimize in the RRM framework can be described as:

$$\mathbf{x}_i^{t+1} = \arg\min_{\mathbf{x}_i} \mathbb{E}_{z_i \sim \mathcal{D}_t} \left[ -\mathbf{z}_i^T \mathbf{x}_i + \frac{\lambda}{2} \|\mathbf{x}_i\|^2 \right] \tag{53}$$

where $\lambda$ is a hyperparameter for the regularization term (= 70 for our experiments). For any player $i$, each element of $\mathbf{x}_i$ is clipped to be between the range of $[-30, 30]$ and the initial price $\mathbf{x_i^0}$ is sampled randomly from a uniform distribution on $[0, 1]$.

**Results.** Figure 5 shows the plot for loss shift due to performativity and the performative risk versus the iterations averaged over 200 runs. For this plot, we assume player 1 is the player who makes the predictions and adjusts to the performative effects introduced due to the actions of player 2. It can be clearly observed that as we increase the aggregation window from $1 \rightarrow 2 \rightarrow 4 \rightarrow t/2 \rightarrow all$, we

[7]Uber and Lyft dataset from Boston, MA, 2019: https://www.kaggle.com/datasets/brllrb/uber-and-lyft-dataset-boston-ma

get mostly lower loss shifts and hence, an improvement in the convergence rate. Since we start at random price value, taking the past into account in the beginning makes the algorithm worse but the effect is neutralized as the data from more time steps is observed. Given the simple linear nature of the problem, this is a significant improvement and provides evidence for our claims about using the data from the previous snapshots. Secondly, performative risk plot in figure 5 also highlights that all methods converge to points having very close values of performative risk, with the methods having larger $\tau$ showing oscillations with smaller amplitude.

# K Proximal RRM Convergence in Perdomo et al. [2020] Framework

**Theorem 9 (Proximal RRM Convergence)** *Suppose the loss $\ell(z; \theta)$ is $\beta$-jointly smooth and $\gamma$-strongly convex. If the distribution map $D(\cdot)$ is $\epsilon$-sensitive, then for*

$$G(\theta) = \arg\min_{\phi} \mathbb{E}_{z \sim D(\theta)}\big[\ell(z; \phi)\big] + \frac{\lambda}{2}\|\theta - \phi\|^2, \tag{54}$$

*we have, for all $\theta, \theta' \in \Theta$,*

$$\|G(\theta) - G(\theta')\| \leq \frac{\epsilon\beta + \lambda}{\gamma + \lambda}\|\theta - \theta'\|.$$

*Furthermore, if $\frac{\epsilon\beta + \lambda}{\gamma + \lambda} < 1$, then $G$ is a contraction, possesses a unique fixed point $\theta_{PS}$, and the proximal RRM iterates $\theta_{t+1} = G(\theta_t)$ converge linearly:*

$$\|\theta_t - \theta_{PS}\| \leq \left(\frac{\epsilon\beta + \lambda}{\gamma + \lambda}\right)^t \|\theta_0 - \theta_{PS}\|.$$

**Proof.** Fix $\theta$ and $\theta'$. Let

$$f(\phi) = \mathbb{E}_{z \sim D(\theta)}\ell(z; \phi) + \frac{\lambda}{2}\|\theta - \phi\|^2, \quad f'(\phi) = \mathbb{E}_{z \sim D(\theta')}\ell(z; \phi) + \frac{\lambda}{2}\|\theta' - \phi\|^2.$$

Because $\ell(z; \phi)$ is $\gamma$-strongly convex, both $f$ and $f'$ are $(\gamma + \lambda)$-strongly convex. Hence

$$f\big(G(\theta)\big) - f\big(G(\theta')\big) \geq \big(G(\theta) - G(\theta')\big)^\top \nabla_\phi f\big(G(\theta')\big) + \frac{\gamma + \lambda}{2}\|G(\theta) - G(\theta')\|^2, \tag{55}$$

$$f\big(G(\theta')\big) - f\big(G(\theta)\big) \geq \big(G(\theta') - G(\theta)\big)^\top \nabla_\phi f\big(G(\theta)\big) + \frac{\gamma + \lambda}{2}\|G(\theta') - G(\theta)\|^2. \tag{56}$$

Since $f$ is minimized at $G(\theta)$, the inner product in 56 is non-negative. Combining 55 and 56 yields

$$\big(G(\theta) - G(\theta')\big)^\top \nabla_\phi f\big(G(\theta')\big) \leq -(\gamma + \lambda)\|G(\theta) - G(\theta')\|^2. \tag{57}$$

Define the regularized loss

$$\ell_\theta(z; \phi) = \ell(z; \phi) + \frac{\lambda}{2}\|\theta - \phi\|^2.$$

The map

$$z \longmapsto \frac{\big(G(\theta) - G(\theta')\big)^\top \nabla_\phi \ell_\theta\big(z; G(\theta')\big)}{\beta\|G(\theta) - G(\theta')\|}$$

is 1-Lipschitz in $z$ because of the $\beta$-joint smoothness of $\ell$. The $\epsilon$-sensitivity of $D(\cdot)$ then implies

$$\sup_{g \text{ is 1-Lip}} \big|\mathbb{E}_{z \sim D(\theta)}g(z) - \mathbb{E}_{z \sim D(\theta')}g(z)\big| \leq \epsilon\|\theta - \theta'\|. \tag{58}$$

Using the 1-Lipschitz function above in 58 gives

$$\left|\frac{(G(\theta) - G(\theta'))^\top}{\beta\|G(\theta) - G(\theta')\|}\big(\mathbb{E}_{z \sim D(\theta)}\nabla_\phi \ell_\theta(z; G(\theta')) - \mathbb{E}_{z \sim D(\theta')}\nabla_\phi \ell_\theta(z; G(\theta'))\big)\right| \leq \epsilon\|\theta - \theta'\|.$$

Unfolding $\ell_\theta$ and rearranging, we obtain

$$-\epsilon\beta\|\theta - \theta'\|\,\|G(\theta) - G(\theta')\| \leq \big(G(\theta) - G(\theta')\big)^\top\big[\nabla_\phi f\big(G(\theta')\big) - \nabla_\phi f'\big(G(\theta')\big)\big]$$
$$+ \lambda\big(G(\theta) - G(\theta')\big)^\top(\theta' - \theta). \tag{59}$$

Since $G(\theta')$ minimizes $f'$, we have $\big(G(\theta) - G(\theta')\big)^\top \nabla_\phi f'\big(G(\theta')\big) \geq 0$. Thus

$$-\epsilon\beta\|\theta - \theta'\|\,\|G(\theta) - G(\theta')\| \leq \big(G(\theta) - G(\theta')\big)^\top \nabla_\phi f(G(\theta'))$$
$$+ \lambda\|G(\theta) - G(\theta')\|\,\|\theta - \theta'\| \tag{60}$$

by the Cauchy–Schwarz inequality. Combining 59 and 60 gives

$$(-\epsilon\beta - \lambda)\|\theta - \theta'\|\,\|G(\theta) - G(\theta')\| \leq \big(G(\theta) - G(\theta')\big)^\top \nabla_\phi f(G(\theta')), \tag{61}$$

and substituting the upper bound 57 for the right-hand side yields

$$(-\epsilon\beta - \lambda)\|\theta - \theta'\|\,\|G(\theta) - G(\theta')\| \leq -(\gamma + \lambda)\|G(\theta) - G(\theta')\|^2. \tag{62}$$

Since $\|G(\theta) - G(\theta')\| \geq 0$, dividing both sides of 62 by $(\gamma + \lambda)\|G(\theta) - G(\theta')\|$ gives

$$\|G(\theta) - G(\theta')\| \leq \frac{\epsilon\beta + \lambda}{\gamma + \lambda}\|\theta - \theta'\|. \tag{63}$$

**Stability of the Proximal RRM Solution in Standard RRM**

We examine whether the fixed point of the Proximal RRM introduced in Theorem 9 coincides with the performatively stable point of RRM.

Assume that $\frac{\epsilon\beta}{\gamma} < 1$, Under this condition, a performatively stable point of RRM exists and it's unique. Because the proximal term vanishes at $\phi = \theta_{PS}$, optimality of $\theta_{PS}$ yields

$$\mathbb{E}_{z\sim\mathcal{D}(\theta_{PS})}\,\ell(z;\theta_{PS}) \;\leq\; \mathbb{E}_{z\sim\mathcal{D}(\theta_{PS})}\,\ell(z;\phi) \;\leq\; \mathbb{E}_{z\sim\mathcal{D}(\theta_{PS})}\,\ell(z;\phi) + \frac{\lambda}{2}\|\theta_{PS} - \phi\|^2 \quad \forall\phi.$$

Hence

$$\theta_{PS} \;=\; \arg\min_{\phi}\; \mathbb{E}_{z\sim\mathcal{D}(\theta_{PS})}\,\ell(z;\phi) + \frac{\lambda}{2}\|\theta_{PS} - \phi\|^2, \tag{64}$$

so $\theta_{PS}$ is also a fixed point of the Proximal RRM.

Note that, for any $\lambda > 0$,

$$\frac{\epsilon\beta}{\gamma} \;\leq\; 1 \;\Longrightarrow\; \frac{\epsilon\beta + \lambda}{\gamma + \lambda} \;\leq\; 1. \tag{65}$$

By Theorem 9, inequality 65 guarantees that the Proximal RRM admits a *unique* fixed point, denoted by $\theta_{PS}^{\lambda}$. Since $\theta_{PS}$ satisfies 64 and the minimiser of 64 is unique, we conclude

$$\theta_{PS}^{\lambda} \;=\; \theta_{PS}.$$

Thus, under $\frac{\epsilon\beta}{\gamma} < 1$, the performatively stable solution of RRM is identical to the unique fixed point of the Proximal RRM.

## L  Limitations

Despite our contributions, several limitations warrant discussion. First, although we prove that ARM achieves performative stability under a broader set of sensitivity conditions than standard RRM, this does not necessarily translate into faster convergence rates. Second, while the underlying techniques can extend to other iterative schemes, such as Repeated Gradient Descent (RGD), and more general gradient-based methods, we do not provide explicit convergence analyses or lower-bound characterizations for these alternatives. Finally, our entire treatment assumes deterministic, non-stochastic setups with exact access to full distributions; we leave the problem of accommodating sampling noise and stochastic gradients for future work.

