# OpenReview forum: "Tight Lower Bounds and Improved Convergence in Performative Prediction"
_NeurIPS.cc/2025/Conference — NeurIPS 2025 poster_

### Official Review · Reviewer_xaNN · 2025-06-06

**Clarity:** 3
**Significance:** 2
**Originality:** 3
**Rating:** 4
**Confidence:** 3

**Summary:**

This paper proposes a new set of assumptions in performative prediction and the corresponding RRM convergence results. Then based on this result, the authors proved the tightness of the convergence speed of [Perdomo et al., 2020] and [Mofakhami et al., 2023]. Next, the authors proposed a new iterative optimization scheme of PP where each time all historical datasets are used (ARM). This brings an improved (but still linear) convergence rate and more relaxed conditions for the convergence of ARM.

**Questions:**

- What are the challenges to prove the tightness of convergence rates?
- Can you compare your Assumption 1 to the Assumption 1 in [Mofakhami et al., 2023]? Will it be significantly more restrictive?
Other questions seen weaknesses.

**Ethical Concerns:**

["NO or VERY MINOR ethics concerns only"]

**Final Justification:**

I encourage the authors to include the intended modifications and highlight the things in Appendix H. Currently I will retain the positive score.

**Limitations:**

yes (in Appendix)

**Quality:**

3

**Strengths And Weaknesses:**

> Strengths

- The theoretical results on tightness analysis make sense to me. I scanned through Appendix D and the proof seems to be plausible.
- The ARM formulation is clear and the proof sketch for Theorem 5 and 6 are clear.
- The writing style of the main paper is easy to follow.

> Weaknesses

- It is better to provide a proof sketch in main paper. Especially for Theorem 2 which is an extension of [Mofakhami et al., 2023], what are the challenges to prove a tight lower bound? Which different proof techniques are used?
- In Theorem 4, the distribution map needs to satisfy $\frac{\epsilon}{C}$ sensitivity and the $C$ is a condition in Assumption 2. But currently I cannot get how large $C$ would be. Can you provide some examples? Afterall, the increased convergence speed is still linear, so the constant $C$ may not be negligible?

> Minor
- In Assumption 2, is that "for all $f_{\theta}, f_{\theta'}$ and $f_{\theta^{*}}$"?
-  The authors may discuss how restrictive their assumptions are and the related work should include some discussions on recent related literature which have relaxed the convexity assumptions of PP (e.g., [1], [2], and also [Li and Wai, 2024] in your related work section).

[1] Cyffers, Edwige, et al. "Optimal Classification under Performative Distribution Shift." The Thirty-eighth Annual Conference on Neural Information Processing Systems (2024).

[2] Zheng, Xue, Tian Xie, Xuwei Tan, Aylin Yener, Xueru Zhang, Ali Payani, and Myungjin Lee. "ProFL: Performative Robust Optimal Federated Learning." arXiv preprint arXiv:2410.18075 (2024).

---

> ### Author Rebuttal · Authors · 2025-07-31
>
> We thank the reviewer for the constructive comments and careful reading of our manuscript. We answer their questions below and provide important clarifications.
>
> > **Q1 and W1: Proof sketch for Theorem 2.**
>
> To our knowledge, no prior lower bounds have been established for this class of algorithms. While our analysis builds on standard tools, the application to this setting is novel, particularly the use of the specific distribution mapping, which has not been previously reported in the literature to our knowledge. Due to space constraints, we were unable to include the full sketch in the main paper, but we will add a high-level overview of the proof in the revised version.
>
> > **W2: Examples and sensitivity constants.**
>
> For the lower bound (Theorem 2), the example we construct satisfies the sensitivity condition with constant $C = 1$. This suffices to show that the proposed improvement can outperform the convergent problem class addressed by standard RRM. Additionally, Appendix H (line 726) presents another example in the RIR setup with $C = \frac{1}{\delta}$ for $\delta < 1$. We will move this calculation to the main text and make the constants more explicit. See also the end of our answer to Q2 for Reviewer 6Knm for another setup of explicit computation of $C$ that we will add in the appendix.
>
> > **Minor1: Assumption 2 notation.**
>
> The reviewer is correct to point out that Assumption 2 should apply to all $f_\theta$,  $f_{\theta'}$, and $f_{\theta^\*}$. We will correct this in the revised version.
>
> > **Minor2: Assumptions restrictivity.**
>
> We agree that a clearer discussion of how restrictive the assumptions are—and how their interpretation has evolved in recent literature—would improve the paper. We will add a paragraph after the assumptions section comparing our conditions to those in related work, including recent efforts that relax strong convexity or sensitivity requirements.
>
> > **Minor2: Related work.**
>
> We appreciate the insightful comment. In the revised version, we will expand the related work section to include recent literature [2, 3].
>
> [2] Cyffers, Edwige, et al. "Optimal Classification under Performative Distribution Shift." The Thirty-eighth Annual Conference on Neural Information Processing Systems (2024).
>
> [3] Zheng, Xue, Tian Xie, Xuwei Tan, Aylin Yener, Xueru Zhang, Ali Payani, and Myungjin Lee. "ProFL: Performative Robust Optimal Federated Learning." arXiv preprint arXiv:2410.18075 (2024).
>
> > **Q2: Comparison to $\epsilon$-sensitivity in [Mofakhami et al., 2023].**
>
> We agree that a clearer discussion of the restrictiveness of our assumptions would improve the paper. In Appendix H, we provide an illustrative example comparing our assumption to the $\epsilon$-sensitivity condition in Mofakhami et al. (2023). In their setup, $\epsilon = \frac{1}{\delta}$, while in ours it becomes $\epsilon = \frac{1}{\delta}\left(1 + \frac{1 - \delta}{2\sqrt{\delta}}\right)$ for $0<\delta<1$. As shown in Theorem 8 and discussed in Appendix H, this still yields a faster convergence result. We will include additional discussion in the revision, clarifying the implications and practical restrictiveness of our assumptions.

---

### Official Review · Reviewer_n4o8 · 2025-07-02

**Clarity:** 3
**Significance:** 3
**Originality:** 3
**Rating:** 5
**Confidence:** 4

**Summary:**

The authors study repeated risk minimization in the context of performative prediction, where a deployed model influencens the data-generating distribution. Repeated risk minimization converges, under certain assumptions, to a stable state at which the chosen model is minimizes risk on the distribution it generates. In other words, the model is a fixed point of the risk minimization operator.

The paper presents tight upper and lower bounds on the the linear convergence of repeated risk minimization under two different sets of assumptions from prior work, one by Mofakhami et al. (2023) and one by Perdomo et al. (2020). Both sets of assumptions require strong convexity of the loss and the property that a small model change can only lightly affect the resulting distribution (called sensitivity of the distribution map). In addition, certain gradient norms have to be small.

The two sets of assumptions differ in how distance in distribution space is measured, and what norms are bounded, leading to incomparable and complementary convergence guarantees.

In addition, the authors study a new class of algorithms, affine risk minimizers, that use previously encountered datasets to speed up convergence.

**Questions:**

Minor comments:

- There may be a typo in Equation 5 in the definition of the subscrip to the norm. Should it be f_\theta or f_{\theta^*}?

- It would be helpful to state the relationship between \chi^2-divergence and Wasserstein distance W1 to make the comparison of the results easier. I understand that Mofakhami et al. (2023) had some discussion about this, but the reader may not know that work. It's worth restating some of it for clarity.

- Along the same lines, it would help to provide some guidance on when each of the two results is preferable.

Questions:

Are you aware of settings where affine risk minimizers converge under strictly weaker assumptions? Even a proof of concept would be interesting, in my opinion.

**Ethical Concerns:**

["NO or VERY MINOR ethics concerns only"]

**Final Justification:**

I maintain my positive evaluation after the discussion period.

**Limitations:**

Yes

**Paper Formatting Concerns:**

Very minor:
- Things like "w.r.t. predictions" should be "w.r.t.~predictions" to avoid excessive horizontal space.

**Quality:**

3

**Strengths And Weaknesses:**

Performative prediction is a widely studied problem and within this area the convergence of retraining has seen much of the attention. The paper presents solid technical results, taking a good step on top of existing work. Conceptually, not too much is new, but the results require a good bit of new technical material. It's to have matching upper and lower bounds for both the Mofakhami assumptions and the Perdomo assumptions.

I like the idea of affine risk minimizers. It's quite natural in practice to mix in data from previous iterations and not just the most recent datasets. This makes a lot of sense. It's nice to see that this leads to improvements, even if the improvements are only in terms of constant factors in the base of the exponential. It would've been even nicer to see some settings where affine risk minimizers achieve weaker assumptions and not just faster convergence.

To summarize, this is a solid technical contribution in the area of performative prediction that merits acceptance.

---

> ### Author Rebuttal · Authors · 2025-07-31
>
> We thank the reviewer for the thoughtful and encouraging feedback. We answer their questions and provide important clarifications below.
>
> > **Minor1: Typo in Eq. 5.**
>
> We clarify that the original notation is correct, though we agree it can be confusing. The purpose of Eq. 5 is to define the norm structure that will be used throughout the paper. While we write $𝑓_{\theta^\*}$, the variable $\theta^\*$ here refers to an arbitrarily chosen reference parameter and is not meant to indicate an optimal solution. We will make this clarification explicit in the revised version to avoid confusion.
>
> > **Minor2: Discussions on relations between $\chi^2$ and $W1$ distance.**
>
> As suggested, we will include a brief discussion in the revision that clarifies the connection between the $\chi^2$ divergence and the Wasserstein distance $W1$. As noted in Remark 1 in Mofakhami et. al  (2023), $\epsilon$-sensitivity w.r.t. $\chi^2$ divergence is a stronger condition than $\epsilon$-sensitivity w.r.t. the Wasserstein distance in cases where the diameter of the input space is small enough. However, since one does not always imply the other, we analyze each framework separately based on its respective assumptions.
>
> > **Minor3: Clarifying assumption roles.**
>
> We also appreciate the suggestion to contextualize our assumptions. In the revised version, we will add a dedicated paragraph comparing the assumptions used in Perdomo et al. (2020) and our adapted version of those in Mofakhami et al. (2023), with emphasis on how these relate to our theoretical bounds.
>
> > **Q1: Convergence under weaker assumptions?**
>
> We agree that it is of interest to identify settings where affine risk minimisers converge under strictly weaker assumptions. We note that Theorem 4 already provides an instance of such a setting: it guarantees convergence under relaxed constants, which are weaker than those of previous results (lines 217-220). Namely, when $1\leq \frac{\sqrt{\epsilon}M}{\gamma} < 1.155$, 2-snapshots ARM converges whereas RRM does not converge. We will make this clearer in the next revision.
>
> > **Formatting concerns.**
>
> Thanks for the suggestion. We’ll make sure to add the non-breaking space (~) in those instances.

---

> > ### Comment · Reviewer_n4o8 · 2025-08-05
> >
> > Thanks for the clarifications and proposed updates. SGTM.

---

### Official Review · Reviewer_6Knm · 2025-07-03

**Clarity:** 3
**Significance:** 3
**Originality:** 3
**Rating:** 4
**Confidence:** 2

**Summary:**

The paper introduces a new class of algorithms for the performative optimization problem. Theoretically, the proposed method not only matches their shown lower bounds but also improves upon the performance of existing Repeated Risk Minimization (RRM) approaches. Numerical experiments are provided to demonstrate the practical effectiveness of the method.

**Questions:**

Q1: In Assumption 2, what is defined as the "initial distribution"?

Q2: Also, for Assumption 2, is this assumption common in the literature of PO or for applications? Could the authors provide some settings satisfying the assumptions?

Q3: How are alpha's in (13) specified in Lemma 1? Will any arbitrary choice of alpha_i^(t) work in this case?

Q4: What if the distribution map is misspecified? Will the algorithm still work given some misspecification error?

**Ethical Concerns:**

["NO or VERY MINOR ethics concerns only"]

**Final Justification:**

Thank you for your notification. I am satisfied with the authors' response, and all my questions are resolved. So I still keep on the positive side.

**Quality:**

3

**Strengths And Weaknesses:**

Strengths:
S1: Introduce a new algorithmic class, ARM, with a better convergence rate than existing RRM.
S2: Theoretically show upper bounds of the proposed algorithm class, matching the provided lower bound and breaking the lower bound for existing RRM.
S3: Numerical experiments are also included to show the performance.

Weaknesses:
W1: It would be better to further interpret the assumptions, especially Assumption 2, including whether they are normal in the literature or some specific examples satisfying the assumptions.

---

> ### Author Rebuttal · Authors · 2025-07-31
>
> We thank the reviewer for the detailed feedback and the positive assessment of our algorithmic, theoretical, and empirical contribution. We answer their questions below.
>
> > **Q1 (What is the “initial distribution” p(x) in Assumption 2?).**
>
> The initial distribution is the base distribution 𝑝(𝑥) introduced in Eq. (8), identical to the reference distribution referred to as the base distribution in Mofakhami et al. (2023) and the baseline distribution in Perdomo et al. (2020). Conceptually,  𝑝(𝑥) corresponds to the pre-deployment or organic exposure distribution—i.e., the distribution of inputs before any model intervention occurs. This notion is also standard in the recommender systems literature: for instance, Schnabel et.al. [1] use the Yahoo! R3 dataset, which provides an unbiased test set where a subset of 5,400 users was each shown 10 randomly selected songs to rate. Since the songs were chosen uniformly at random, the resulting test data is Missing Completely At Random (MCAR), offering an intervention-free ground truth. We will clarify this in the final version.
>
> [1] Schnabel, Tobias, Adith Swaminathan, Ashudeep Singh, Navin Chandak, and Thorsten Joachims. "Recommendations as Treatments: Debiasing Learning and Evaluation." ICML 2015.
>
>
> > **Q2 (realism of Assumption 2 and discussion on assumptions).**
>
> We will include a dedicated paragraph in the revision discussing the assumptions in greater depth. All four assumptions are directly inherited or adapted from prior work in performative prediction—specifically, Mofakhami et al. (2023) and Perdomo et al. (2020). In particular, Assumption 2 is structurally identical to the norm-equivalence condition introduced by Mofakhami et al., who also derive explicit expressions for the constants in our RIR experimental setup: namely, $C=\frac{1}{\delta}$ and $c=\frac{1}{2-\delta}$ for $0<\delta<1$. This connection is currently mentioned briefly in Appendix H (line 726), and we will expand on it in the final version. We will also include additional discussion in the appendix outlining some conditions under which Assumption 2 holds in practice. In particular, it holds if the possible densities have common supports with a lower and upper bounded ratio; then we can use something like $c:=\inf_{\theta, z \in support} \frac{p_{f_{\theta}}(z)}{p(z)}$ and $C:=\sup_{\theta, z \in support} \frac{p_{f_{\theta}}(z)}{p(z)}$.
>
> > **Q3 (choice of alpha in Lemma 1).**
>
> We use uniform weights over the final two iterations $\alpha^{(t)}_{t-1}=\alpha^{(t)}_t=12$ (Eq.14).
>
> > **Q4 (What if distribution map is misspecified?).**
>
> Our analysis focuses on RRM and ARM, which do not query the distribution map \$D(\cdot)\$ but instead rely on empirical datasets collected post-deployment, assuming access to an (infinite) sample. While we do not address the misspecification from finite sample sizes in this work—particularly for ARM—this issue has been previously analyzed in the context of RRM by Perdomo et al. (2020).
>
>
> We hope these clarifications fully resolve the reviewer’s concerns, and thank you again for the constructive review.

---

### Decision · Program_Chairs · 2025-09-17

**Decision:**

Accept (poster)

**Comment:**

This paper considers the performative prediction problem and studies an improved RRM convergence results over [Mofakhami et al., 2023]. The new results consist of giving a laxer criterion for linear convergence and tight characterization of the convergence rate. Moreover, an affine risk minimizer is proposed which enjoys further improved rates.

The initial reviews of the paper is quite positive, where the reviewers found the submission to propose a new class of algorithm based on the affine risk minimizer that is able to achieve better rate than traditional performative prediction works. There are no particular disadvantage for the paper and it has therefore passed the acceptance threshold. The authors are reminded to take the comments into account when preparing the camera ready version.